# Spuriosity Rankings:
# Sorting Data to Measure and Mitigate Biases

**Mazda Moayeri**[1]  **Wenxiao Wang**[1]  **Sahil Singla**[2,1*]  **Soheil Feizi**[1]

[1] University of Maryland  [2] Google

{**mmoayeri**, wwx, sfeizi} @umd.edu,  sasingla@google.com

←Low Spuriosity  Class: *Otter*  High Spuriosity→

Class: *Rhinoceros Beetle*

Class: *Lighter*

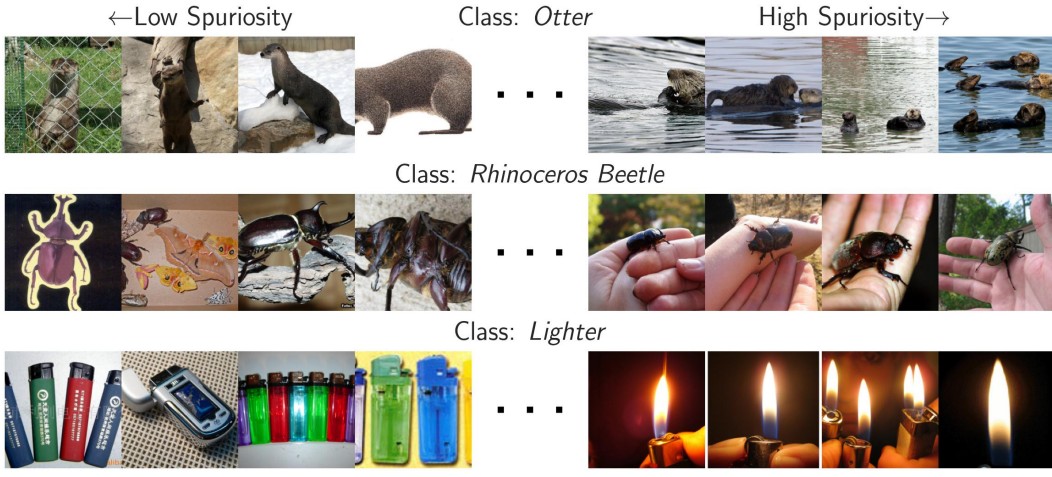

Figure 1: Spuriosity rankings sort images based on the presence of spurious cues. The rankings reveal hidden spurious features that models rely on, like *rippling water, human hands,* and *flame in the dark* (right; top to bottom), and uncover minority subpopulations where spurious cues are absent (left).

## Abstract

We present a simple but effective method to measure and mitigate model biases caused by reliance on spurious cues. Instead of requiring costly changes to one's data or model training, our method better utilizes the data one already has by sorting them. Specifically, we rank images within their classes based on spuriosity (the degree to which common spurious cues are present), proxied via deep neural features of an interpretable network. With spuriosity rankings, it is easy to identify minority subpopulations (i.e. low spuriosity images) and assess model bias as the gap in accuracy between high and low spuriosity images. One can even efficiently remove a model's bias at little cost to accuracy by finetuning its classification head on low spuriosity images, resulting in fairer treatment of samples regardless of spuriosity. We demonstrate our method on ImageNet, annotating 5000 class-feature dependencies (630 of which we find to be spurious) and generating a dataset of $325k$ soft segmentations for these features along the way. Having computed spuriosity rankings via the identified spurious neural features, we assess biases for 89 diverse models and find that class-wise biases are highly correlated across models. Our results suggest that model bias due to spurious feature reliance is influenced far more by what the model is trained on than how it is trained.

---

*Work carried out while at the University of Maryland.

37th Conference on Neural Information Processing Systems (NeurIPS 2023).

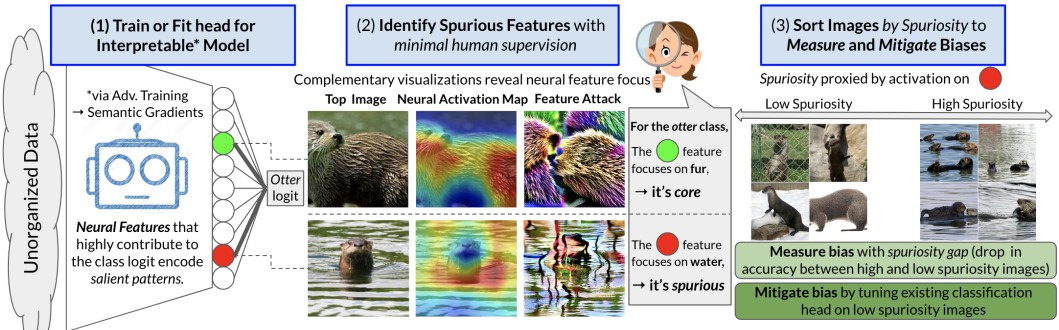

Figure 2: Overview of our framework. Per class, we sort images by *spuriosity*, which we proxy via the activation of neural features (i.e. nodes in the penultimate layer of a trained classifier) that focus on spurious cues. After sorting, measuring model bias to the discovered spurious features is as simple as computing the drop in accuracy across high and low spuriosity images. This bias can be mitigated efficiently by tuning the classification head on low spuriosity images.

# 1   Introduction

Deep models often exhibit bias, underperforming on certain minority subpopulations [6, 10]. One cause of bias is the tendency of deep models to rely on spurious features [45, 39, 1], as samples that retain spurious correlations are predicted more accurately than rarer samples where spurious correlations are broken. For example, a classifier may learn to associate water with the *otter* class if many training images contain water, which in turn can lead to reduced accuracy on images of otters out of water. Spurious feature reliance also hinders robust generalization (e.g. medical systems trained in one hospital may fail when deployed to others [57, 12]), potentially posing major reliability risks.

Recent trends suggest that scaling up models and datasets may resolve many existing limitations [50]. However, in addition to being costly, collecting more data would not alter the long-tail data distribution that leads to spurious feature reliance, as the spurious cues will remain just as correlated, if not more, with class labels. Instead of seeking more data, we suggest to *better utilize the data one already has*. Specifically, we aim to rank images within their classes by *spuriosity*, which can be understood informally as the degree to which relevant spurious cues are present. Figure 1 shows examples of our rankings. For the *otter* class, high spuriosity images prominently contain the common spurious feature of water, while low spuriosity images do not, instead displaying the rarer case of otters on land.

Spuriosity, while intuitive, is rather challenging to measure scalably: for each class, one must (i) discover relevant cues (i.e. those that a model will rely upon), (ii) identify the spurious ones, and (iii) obtain a function that scores the level to which these spurious cues are present in a given image. To this end, we propose to extract spurious concept detectors from within an *interpretable model* trained on the given data. Namely, we automatically select neural features (i.e. nodes in the penultimate layer) that contribute highly to a class logit, as they encode concepts that are both salient to that class and relied upon by the model. Then, we interpret the concept each neural feature detects, with limited human supervision, so to keep the ones activated by spurious cues. Finally, we average the activation on these neural features to yield a scalar for each image that proxies the presence of relevant spurious cues (i.e. spuriosity), thus enabling sorting. Once data is ranked by spuriosity, multiple significant benefits emerge immediately: First, low spuriosity images **reveal minority subpopulations**, where common spurious correlations are broken. Second, we can easily **quantify model bias** as the drop in accuracy between high and low spuriosity images; we call this *spurious gap*. Third, we can **mitigate bias** towards high spuriosity images by simply finetuning the classification head on low spuriosity images.

Figure 2 diagrams our approach. The interpretable model we use to detect relevant neural features and extract spurious ones (steps 1 and 2 respectively) is adversarially trained, as in [46]. However, we show for the first time that we can extend this framework *without adversarial training*. Namely, to discover patterns salient for a new task, it suffices to merely fit a linear classification head on the new data over fixed adversarially pre-trained neural features. This dramatically increases the efficiency of the method, and expands its use cases to low-data regimes where adversarial training is less feasible. We carry out steps 1 and 2 at an unprecedented scale, resulting in $5000$ annotated class-neural feature

dependencies over all of the ubiquitous ImageNet, including 630 spurious features discovered over 357 classes. For these classes, we compute spuriosity rankings (step 3), which, along with a web-UI[*] for easy viewing and $325k$ soft segmentations of salient visual features in ImageNet, we present as a dataset to shed insight on *how* deep models perform ImageNet classification.

Using spuriosity rankings, we compute *spurious gaps* (drop in accuracy from high to low spuriosity images) for **89 models**, observing that all models underperform on low spuriosity images. Models exhibiting the most and least bias to spurious features are CLIP and adversarially trained models respectively, corroborating prior work [28, 37] and thus further validating our rankings. We also observe diverse models to have highly correlated class-wise spurious gaps. This implies that models absorb the same biases, suggesting that *bias is influenced far less by how a model is trained, and far more by what it is trained on*. Thus, frameworks like ours that demystify large datasets can be instrumental in understanding and mitigating bias. Indeed, we show that finetuning existing classification heads on low spuriosity images is an efficient and effective model-agnostic way to close spurious gaps at little cost of validation accuracy, resulting in fairer, more stable treatment of inputs, regardless of spuriosity.

We also observe surprising instances of negative spurious gaps, i.e. classes where models perform worse when spurious cues are present. These cases are due to spurious feature collision, where one class' spurious feature is also correlated more strongly with another class. Existing benchmarks [23] lack the scale of ours to reveal this atypical yet harmful consequence of spurious feature reliance. Closer inspection reveals high spuriosity images for classes with negative spurious gaps are often mislabeled or contain multiple objects. Thus, spuriosity rankings can help in flagging label noise, and in some cases (via cropping to isolate regions activating core neural features), resolving it.

In summary, we present a simple, extensible method for (i) discovering spurious features (ii) measuring and (iii) mitigating model biases caused by these spurious features. Centering on the fact that spurious correlations are determined by data, our solution focuses on better understanding and using the data one already has, in contrast to data-agnostic approaches which focus on altering model training [2, 41]. Arguably, our method provides more practical robustness, as we first interpret potential biases a model trained on the given data is likely to suffer, and then devise solutions specifically tailored to these biases. By shifting focus from model training to data understanding, spuriosity rankings allow for efficient and interpretable robustness interventions, towards more reliable AI.

## 2   Review of Literature

**Spurious correlation benchmarks** typically consider a small set of predefined spurious attributes fixed across classes [55, 42, 25], and real-world benchmarks tend to be very specific to a narrow task [23]. In both cases, the fixed spurious attributes are annotated manually. Similarly, many have proposed variants of ImageNet's validation set, collecting new data (e.g. with altered renditions, as sketches, in household settings and odd poses, etc [16, 49, 3]) and inspecting the drop in accuracy compared to the original validation set. In contrast, our framework provides a *general* method to uncover and automatically annotate (i.e. per sample) the presence of spurious features relevant to *any* task given its training data, as well as measure and mitigate model bias caused by these spurious features. Since bias arises from data, we believe our step of first interpreting the correlations within a dataset, instead of simply choosing spurious features of interest apriori, is paramount in order to build *practical* robustness (i.e. to distribution shifts that are naturally occurring and potentially harmful, since they break spurious correlations a model trained on the given data likely uses). Also, we account for the fact that whether or not a feature is spurious depends crucially on its class: e.g. while the *water* feature is spurious for the *otter* class, it is not spurious for a class like *lakeshore* (also, see figure 3). Most benchmarks overlook this, defining spurious features class-agnostically. Other approaches to benchmark spurious feature reliance require synthetic data [2, 54, 13] or corrupting spurious regions, which undesirably takes models out of distribution [29, 30]. Instead, our benchmark operates only on natural images one already has, using spuriosity to curate subsets with spurious cues strongly present and absent.

**Natural image-based interpretability** consists of explaining inferences using examples from the corpus [8, 11] or generated examples intended to visually reflect a traversal between concepts [15]. Similarly, our rankings traverse from strong spuriosity to minority examples with spurious cues absent, though we do not require generative models. Other related approaches are influence functions [22] and data models [19], though they are computationally expensive and the former can be fragile [4].

---

[*]`salient-imagenet.cs.umd.edu`

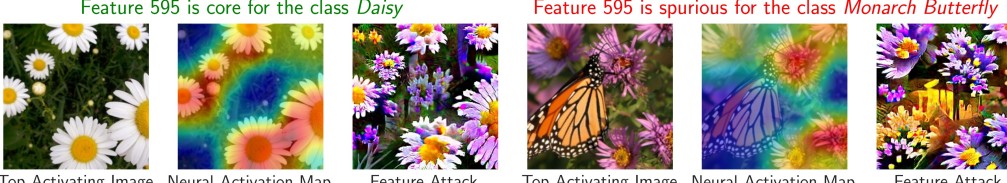

Figure 3: Examples of the visualizations we use to identify the focus of a robust neural feature. Here, feature 595 focuses on flowers, making it core for *Daisy* and spurious for *Monarch Butterfly*.

**Mitigating spurious correlation reliance** often involves altering the training objective to learn more invariant features [2, 38] or directly optimize worst group accuracy [41, 18, 40, 58, 48]. Most methods require extra supervision with respect to non-class-label attributes [33]. Retraining approaches either focus on correcting errors [24, 59, 32] or using balanced data (i.e. dominant spurious cues balanced with minority samples) [20]; our rankings would directly assist in constructing such balanced subsets. We more directly compare error-correcting approaches to our method in Appendix F.

**Neural features** have been studied to interpret models [35, 17]. Notably, adversarially robust neural features have enhanced interpretability [47, 52]. Singla and Feizi [46] introduced a framework that uses robust neural features to discover core and spurious data patterns with minimal human supervision. We apply this framework to scalably compute spuriosity rankings. In the next section, we explain this framework and our novel contributions to it, including how to use the framework without requiring its most prohibitive step of adversarial training, thus dramatically increasing its use cases.

## 3    Discovering Spurious Features in ImageNet and Beyond

A key component of our framework for interpreting and improving a model's robustness to spurious correlations is that we first discover relevant spurious features *based on the training data*, as opposed to data-agnostic approaches. Specifically, we leverage the feature discovery method of [46], performing it at an unprecedented scale (i.e. across all of ImageNet). We now provide a brief overview of the method, details on our expansion, and a novel extension of the method with significant impacts. For complete details on feature discovery and our human studies, we refer readers to Appendix I.

### 3.1    5000 **Reasons Deep Models Use to Perform ImageNet Classification**

Despite its ubiquity, ImageNet (and any other large-scale dataset) is opaque in the sense that a human cannot anticipate the patterns a model trained on it will associate with each class. The reason for this is simple: humans cannot process a million (or even 1000) images at once. Moreover, the patterns a human may use will not necessarily align with those a model will use [14]. Nonetheless, understanding the features (especially the spurious ones) that a model will rely upon is instrumental in anticipating and mitigating the biases a model will suffer from.

To understand the features any general model may rely upon, we inspect the *neural* features of a single, interpretable model; namely, an *adversarially trained* one. Adversarial training leads to perceptually aligned gradients [44], which greatly improve the utility of gradient-based interpretations. Specifically, using the gradient of a neural feature's activation w.r.t. the input image, one can reliably generate a heatmap highlighting the input regions activating the neural feature, and even perturb the input image to visually amplify the cue that the feature detects; the latter method is called a feature attack. Thus, given a robust neural feature, a human can annotate its function as core or spurious for a class by inspecting the images within the class that activate the feature most (we use top 5), along with heatmaps and feature attacks for those images (see Figure 3). Note that given a class, one can automatically select important neural features based on the average contribution of the feature to the class logit, which can easily be computed by inspecting feature activations and linear classification head weights. These steps make up the feature discovery framework of [46]; to summarize, (i) adversarially train a model, (ii) automatically select important neural features per class, and (iii) use complementary visualization techniques to annotate a neural feature as core or spurious with minimal human supervision.

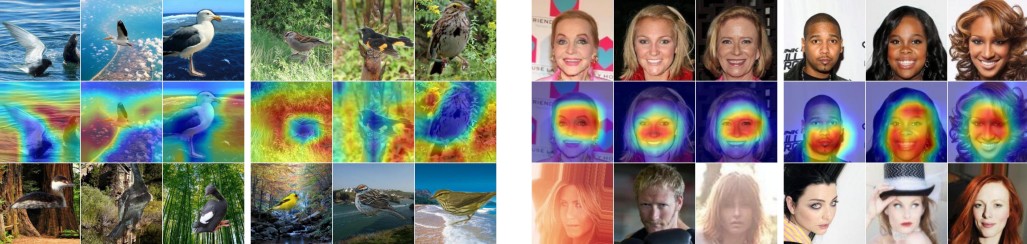

(a) **Background bias** in Waterbird vs. Landbird Classifi- (b) **Racial bias** in Celeb-A Blond vs. Brown Hair Classification; the focus should be on the bird, not background. sification; the focus should be on the hair, not skin.

Figure 4: With only linear layer retraining, we detect designated spurious attributes like background in the Waterbirds benchmark (left), as well as new spurious dependencies such as racial bias in Celeb-A Hair Classification (right). Presented above are highest activating images (top row), corresponding heatmaps (middle), and lowest activating images (bottom) for spurious features observed for the classes *waterbird, landbird, blond hair,* and *brown hair* (left to right).

Singla and Feizi [46] annotated the 5 most relevant neural features for 232 classes of ImageNet. Through a large-scale human study (details in Appendix I), we expand the analysis to all 1000 classes, resulting in 5000 annotated class-feature pairs, of which 630 are spurious over 357 classes. We host a web-UI to view all 5000 pairs, offering a direct visual look into the patterns a neural network sees and uses across ImageNet. We also verify that heatmaps for annotated features localize the same cue in images whose activation on the feature is in the top $20^{th}$ percentile for the class, successfully validating 95.3% of annotated class-feature pairs (see appendix for details). Thus, we generate 325,000 feature soft segmentations across ImageNet as a bonus. Crucially, the validation confirms that sorting by feature activation is effective in gathering instances where a feature is present.

While we use the feature discovery method of [46] directly, we make a number of impactful contributions atop it. Namely, we expand its use cases significantly by removing the requirement of adversarial training. Also, we overhaul the original procedure for assessing model reliance on spurious features, making it far more efficient, less biased, and more stable (Appendix I.4). Key to our improvements is the use of *lowest* activating images as natural counterfactuals to better interpret spurious features and measure their effects, enabling new cross-class and cross-model analyses. Lowest activating images (never used in [46]) are also crucial for computing and closing spurious gaps (Sections 4.2 and 4.3).

## 3.2 Spurious Feature Discovery *without* Adversarial Training

Adversarially robust neural features are at the heart of our framework, though adversarial training can be challenging, particularly when data is limited. We propose to simply fit a linear layer for a new task atop fixed features from an adversarially robust feature encoder pretrained on ImageNet; prior work shows transfer learning on robust features is very effective [43]. We then perform the usual process of identifying important robust neural features per class and annotating them as core or spurious.

We demonstrate this light-weight extension of the feature discovery framework of [46] on two spurious correlation benchmarks: Waterbirds, and Celeb-A Hair Color classification. Figure 4 shows some of the discovered cues. For Waterbirds, we find neural features that focus entirely on backgrounds (the intended spurious feature for this benchmark). The lowest-ranked images within a class for these features contain the opposite background (land instead of water), thus revealing the minority subgroup (waterbirds on land). For Celeb-A, we observe features that spuriously focus on faces (instead of hair). Further, race appears to be homogeneous amongst the top-ranked images, and opposite in the bottom-ranked images. Thus, a model may spuriously associate brown skin with the brown hair class, likely leading to failures for blond-haired brown-skinned people. To our knowledge, *we are the first to uncover this racial bias in the Celeb-A benchmark*, whose intended spurious feature was gender.

To showcase the ease-of-use of our method and the wide-reaching nature of the spurious correlation problem, we apply the lightweight feature discovery framework to a *non*-spurious correlation benchmark, finding numerous spurious cues. We take UTKFace [60], a dataset of face images, and focus on the tasks of gender and race classification (details in Appendix E). Figure 5 visualizes three detected spurious features: suit and tie for the *male* class, the color pink for the *female* class, and glasses

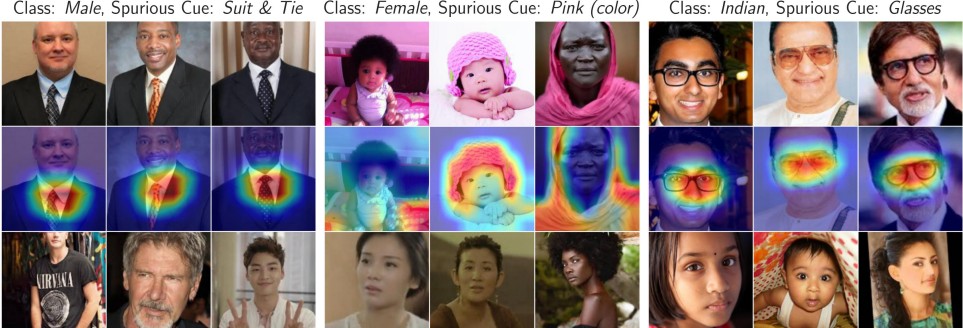

Class: *Male*, Spurious Cue: *Suit & Tie*    Class: *Female*, Spurious Cue: *Pink (color)*    Class: *Indian*, Spurious Cue: *Glasses*

Figure 5: Our lightweight extension of [46] also reveals spurious features in *non*-spurious correlation benchmarks, such as UTKFace gender and race classification. Spurious cues are absent in the lowest activating images (bottom row). Thus, sorting by feature activation uncovers minority subpopulations.

for the *Indian* class. Thus, using robust neural features to discover spurious features is effective, even when the neural features were not trained on the data of interest; simply fitting a linear layer (which can be done in minutes) over these fixed features sufficed in revealing spurious features. The spurious features are prominently displayed in the top activating images, highlighted in the heatmaps, and *notably absent in the least activating images*, indicating that sorting images based on robust neural feature activations can reveal minority subpopulations where spurious correlations are broken. These low spuriosity images prove to be instrumental in measuring and mitigating biases (Section 4).

### 3.3 Is a Human in the Loop Necessary?

Our framework involves a human in the loop, who is given (i) concise insight to the cues a model trained on the given data may rely upon and (ii) agency to decide which of these cues model performance should be invariant to. We believe this increased transparency fosters greater trust with practitioner. Further, the human involvement is limited to viewing a handful of images per class, making our framework tractable (as we demonstrate by performing it on the 1000 class ImageNet dataset). Nonetheless, in cases where minimizing costs is preferred, the human in the loop can be removed, either by automating the feature interpretation step, or by using open-vocabulary models to directly embed concepts into representation space from text descriptions [37, 31]; we describe these variations to our framework in Appendix G. The key idea of our work is to leverage the interpretability of certain existing models to uncover vectors in representation space corresponding to spurious cues, with which we can scalably sort data by computing similarity to these vectors. The method we present uses axis-aligned vectors of adversarially trained network (i.e. robust neural features), though our framework is not restricted to this instance. We now show how sorting enables for greater utilization of the data one already has, towards resolving the biases caused by relying on spurious correlations.

## 4 Measuring and Mitigating Biases with Spuriosity Rankings

### 4.1 Spuriosity Rankings: Organizing Image Data via Robust Neural Features

Equipped with a method to identify robust neural features that detect relevant spurious cues, we now utilize these features to scalably quantify *spuriosity* (i.e. how strongly spurious cues are present in an image). Letting $\mathbf{r}_i(x)$ denote the activation of image $x$ on the $i^{th}$ robust neural feature, and $\mathcal{S}(c)$ denote the set of neural features annotated as spurious for class $c$, we compute spuriosity as follows:

*The **Spuriosity** of an image $x$ for class $c$, with $\mu_{ic}$ and $\sigma_{ic}$ denoting the mean and standard deviation of activations on feature $i$ over class $c$, is approximated as $\frac{1}{|\mathcal{S}(c)|} \sum_{i \in \mathcal{S}(c)} \frac{\mathbf{r}_i(x) - \mu_{ic}}{\sigma_{ic}}$.*

Essentially, we use the activation of robust neural features as spurious concept detectors to proxy the degree to which relevant spurious cues are present in each image of a dataset *efficiently*; that is, with only a single forward pass of the dataset through the adversarially trained network. Given a class, averaging the distribution-adjusted activations of relevant spurious neural features yields a single scalar value for each image, with which images can be ranked within their classes.

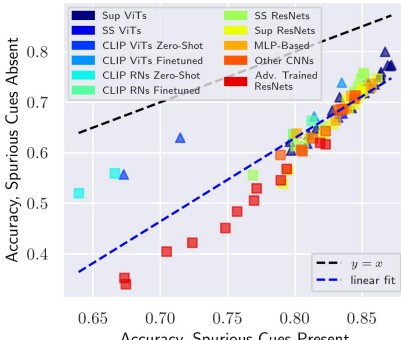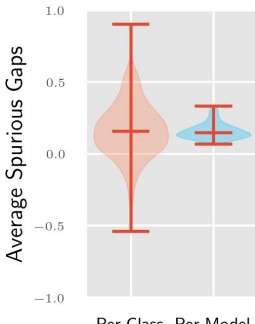

Figure 6: All models are biased: accuracy is 10 to 30% lower when spurious cues are absent compared to when they are present (**left**). However, bias varies far more significantly across classes than across models (**right**), emphasizing that bias due to relying on spurious cues is crucially class-dependent.

Note that we define the spuriosity of an image *for a specific class*, since whether a neural feature is relevant or spurious depends on what class it is being used for: figure 3 shows an example feature (flowers) that is core for one class (*Daisy*) and spurious for another (*Monarch Butterfly*). Accounting for the crucial class-dependence of the spurious correlation problem is unique to our framework.

## 4.2  Measuring Bias: Computing Spurious Gaps

Relying on spurious cues can bias a model, as its performance may degrade when spurious correlations are broken. Measuring this bias can be expensive, as it requires collecting new data where spurious cues are absent. With spuriosity rankings, we instead can easily select such a subset *from one's own data*. Indeed, high spuriosity images prominently display spurious cues, while low spuriosity images do not (Figure 1). Thus, a natural metric enabled by spuriosity rankings is **spurious gap**, defined as the drop in accuracy between the top-$k$ highest and lowest spuriosity validation images per class. We evaluate spurious gaps over the 357 ImageNet classes with at least one spurious feature discovered.

Figure 6 (left) shows accuracy on the $k = 10$ lowest ($y$-axis) vs. highest ($x$-axis) spuriosity images for 89 diverse models pretrained on ImageNet [*] (details in appendix). We observe *all* models to be biased, suffering lower accuracy when spurious cues are absent. As in [27], we observe a strong correlation between the two accuracy measures. However, some models stray from this linear trend, exhibiting 'effective robustness' by performing significantly better on images with spurious cues absent than the linear trend predicts (e.g. zero-shot CLIP) or the opposite (e.g. adversarially trained models). Prior work on distributional robustness corroborates our findings on the effective robustness (and lack thereof) of CLIP and adversarially trained models respectively [37, 28], thus validating our metric. Essentially, **spuriosity rankings organize our data so that we can *simulate distribution shifts* caused by breaking spurious correlations, so to assess a model's robustness without needing to collect new data**.

We now turn our attention to class-wise spurious gaps (i.e. we average spurious gaps per class over all models, instead of averaging over all classes per model, as done in Figure 6, left). We find that spurious gap varies significantly more across classes than across models (Figure 6, right). Specifically, the sample variance is $3.1\times$ larger for spurious gaps across classes than across models, suggesting that the spurious correlation problem pertains more to the data a model is trained on than to the specifics of the model itself. That is, we argue that class-wise spurious gaps vary so much because each class has its own set of spurious cues that are correlated with the class label to different degrees. In contrast, despite differences in training procedure and architecture, **all models learn from the same data, and thus, absorb the same biases**. To investigate this claim, for each pair of models in our study, we compute the correlation between class-wise spurious gaps. On average, we observe a high Pearson's $r$ of $0.69$.

Figure 7 visualizes this strong correlation for five diverse models compared to the interpretable model used to discover spurious features in Section 3. We highlight this case as it justifies our approach in using a single model (namely, an $\ell_2$ adversarially trained ResNet50) to reveal biases for any general

---

[*]except for the CLIP models, which are trained on more data, and also, perhaps not surprisingly, happen to be the most exceptional in their robustness.

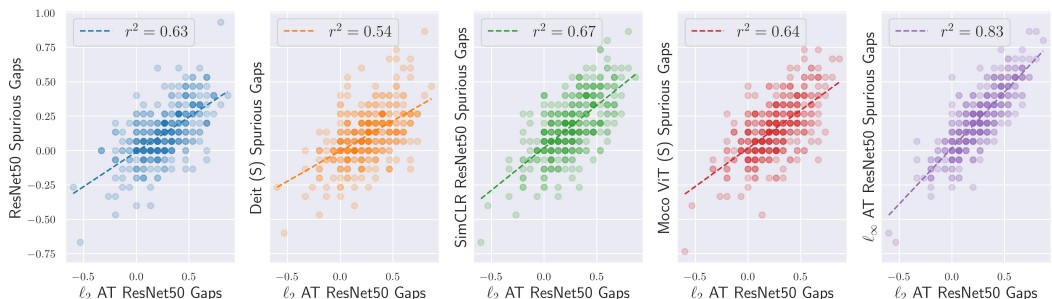

Figure 7: Class-wise spurious gaps are strongly correlated across models. While spuriosity rankings are empirically computed with the help of a single $\ell_2$ adversarially trained (AT) ResNet, diverse models show similar sensitivities to the detected spurious features.

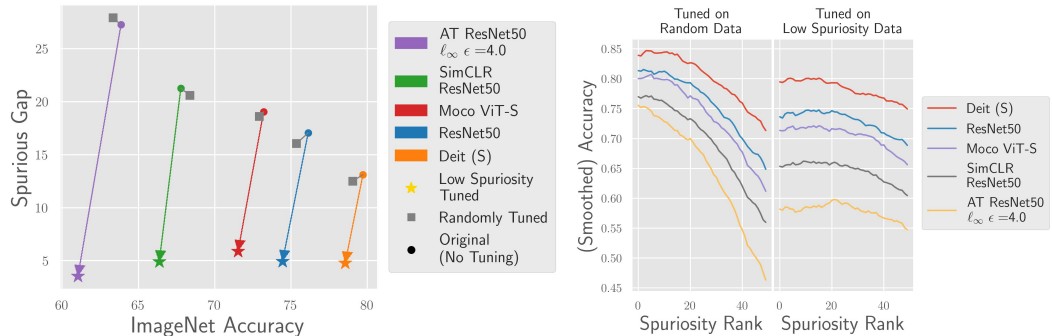

Figure 8: (**Left**) Finetuning on low spuriosity images closes reduces spurious gap to below $5\%$ at little cost of validation accuracy. (**Right**) Closing spurious gap removes bias to images with high spurious ranks; that is, regardless of spuriosity, the model performs with roughly the same accuracy.

model. Because the spurious correlations a model learns depends far more on *what* it is trained on (i.e. the data) than *how* it is trained, we can generalize the spurious features we discover with one model to others. The results of our large-scale empirical analysis support this claim, as all models display bias to the spurious cues discovered with our interpretable model, and furthermore, the degree to which these models are biased per-class are strongly correlated. While it is not surprising that data determines the spurious features a model learns to rely on, we stress this point, as many existing methods for spurious correlation robustness are data-agnostic. In contrast, our method is *data-centric*. Namely, we first discover the spurious features models may rely upon *based on the data*, via an interpretable (essentially, a surrogate) model trained or fit on the data. Then, with spuriosity rankings, we organize our data to allow for efficient measurement of biases per class. We now show how wisely selecting data, via our rankings, for further tuning can mitigate model bias caused by spurious features.

## 4.3 Mitigating Bias: Closing Spurious Gaps

Having identified a bias across ImageNet-trained models, where images that lack spurious cues common to their class are predicted with considerably lower accuracy, we now present a very simple and efficient manner to mitigate this bias at little cost of validation accuracy. In general, to improve some pretrained models on a more specific downstream data distribution, one simply finetunes the model on new data from the target distribution. With spuriosity rankings, we remove the need to acquire new data, as we instead simply select from the training data we already have. In this case, we seek to improve the performance of models on images with spurious cues absent; *this is precisely the low spuriosity images*. Applying our rankings to training data, we can extract a small subset that captures minority subpopulations under-represented in the overall training distribution. Then, we tune the existing classification head on this small subset so to recalibrate the model towards reduced reliance on spurious features and fairer treatment of samples, regardless of spuriosity.

High Spuriosity Images for Classes
with Negative Spurious Gaps

Core Feature Soft Segmentations (via NAMs)

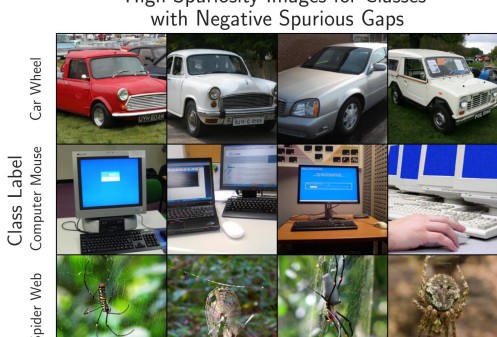
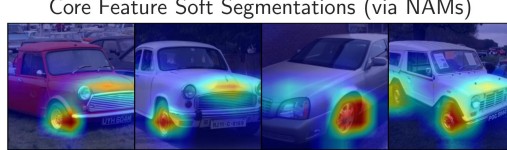

Core Cropped Images

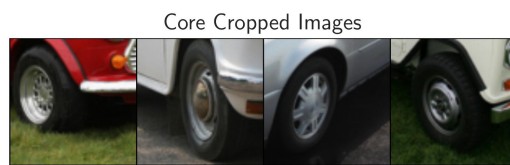

Figure 9: (**Left**) Training images with highest spuriosity for the classes with the most negative spurious gaps. These images often contain multiple objects, at times more prominently showing an object from a class different than the image's label (e.g. *car*, *monitor* or *keyboard*, *spider*). (**Right**) Using neural activation maps for neural features annotated as *core* (i.e. not spurious) for the class may serve as an automated way to crop out the conflicting objects.

We choose to only modify the classification head, as prior work suggests deep models already learn features that are sufficiently informative to correctly predict minority subpopulations [20]. Also, this way, we minimally change the existing model, leading to better retention of validation accuracy, and *extremely fast* bias mitigation, as we only optimize a tiny fraction of the total model parameters using a small fraction of the entire dataset. Specifically, we tune the classification heads of five models spanning diverse architectures (ResNet and ViT) and training procedures (supervised, self-supervised, adversarial) on the 100 lowest spuriosity images for each of the 357 classes for which we discover one spurious feature. This tuning occurs *in minutes* on a single GPU, especially since the data passes through the entire model only once; after caching features, all computation is limited to the linear classification head. Also, we employ early stopping, halting tuning once spurious gap drops below 5%, so to avoid overfitting to the minority subpopulations at the cost of overall accuracy.

Figure 8 visualizes the results of our bias mitigation, plotting the spurious gap (over all 357 classes) vs. validation accuracy before and after our tuning. We also include as a baseline the results for tuning on a subset consisting of 100 randomly selected images from each of the same 357 classes. We find that **low spuriosity tuning mitigates bias at a minimal cost to validation accuracy**, while the baseline does not. Namely, low spuriosity tuning closes spurious gaps, reducing it by 10 to 20%, while only sacrificing 1 to 3% validation accuracy. In the subplot on the right, we see how low spuriosity tuning effectively removes bias to high spuriosity images. While the randomly tuned baseline (like the original model) has far better accuracy on images with high spuriosity ranks, the low spuriosity tuned model has far more stable treatment of all samples, evidenced by a much flatter curve when plotting accuracy vs. spuriosity rank. In appendix F, we consider a second baseline where we tune on samples misclassified by the original model, inspired by existing spurious correlation mitigation techniques [24, 59, 32]. Unlike low-spuriosity tuning, this baseline also fails to close spurious gaps, and actually results in a more substantial drop in overall validation accuracy.

## 5 Additional Application: Flagging and Fixing Labeling Errors

We now demonstrate an additional manner in which spuriosity rankings can help manage and improve the data one already has. Modern datasets, like models, are opaque in the sense that it can be challenging to answer simple yet significant questions about them, largely due to their ever-increasing size. One such question is what spurious features a model will learn to rely upon, and to what degree, after training on a given dataset; our spuriosity rankings framework is designed to answer this question. A related question is how accurate the labels in a dataset are. Even in some of the most widely used vision datasets like ImageNet, label noise has been observed to be pervasive [5], and when it is resolved, models trained on the refined data experience improved accuracy and robustness [56].

A common culprit for label noise in ImageNet is the presence of multiple objects in the same image. Similarly, co-occurring objects are amongst the most common types of spurious cues we observe

models to rely upon. For example, car wheels almost always co-occur with a car, and so, a model may associate the metallic body of a car with the *car wheel* class. Indeed, we discover this spurious feature in our analysis. However, while we typically expect a model to underperform when a spurious cue is absent, we observe the opposite: the car wheel class has a *negative* spurious gap, with models on average having $54.2\%$ lower accuracy on high spuriosity car wheel images than low spuriosity ones. While uncommon, negative spurious gaps are still prevalent, occurring in $15.7\%$ of classes we study. This highlights a second, often overlooked, harm incurred by spurious feature reliance. Namely, since spurious features are not essential to a class, they can be correlated with multiple classes at once, leading to *spurious feature collision*. Indeed, a spurious feature for one class may also be core for another, like the car body being spurious for the *car wheel* class and core for any of the numerous car classes. This is actually the case for a majority of the spurious features we discover, with a staggering $63.8\%$ of them also being core for a different class. In these instances, if a feature $f$ is correlated more with class $c_1$ than $c_2$, high spuriosity images from $c_2$ may be misclassified to $c_1$.

Taking a closer look at classes with severely negative spurious gaps reveals instances of mislabeled training data. Figure 9 shows high spuriosity images for classes with the most negative spurious gaps. These images often more prominently display co-occurring objects different from the class label (such as a car, monitor, or spider) which themselves pertain to another class. Thus, one can automatically flag potential instances of label noise by inspecting high spuriosity images for classes with negative spurious gaps. To quantitatively validate this claim, we leverage ImageNet ReaL labels [5], which indicate for each ImageNet validation image whether objects from multiple classes are present. ReaL labels show $14\%$ of validation images contain multiple objects. We find that for classes with the 5 most negative spurious gaps (averaged over our model suite), high spuriosity (i.e. ranked in top $10\%$ of spuriosity) validation images contain multiple objects in $80\%$ of cases. In contrast, low (bottom $10\%$) spuriosity images from the same classes contain multiple objects in only $8\%$ of cases, indicating that the label noise is likely a result of the hypothesized spurious feature collision. Similarly, for classes who's spurious gap is less than $-20\%$, ReaL labels reveal $42\%$ of the high spuriosity images to contain multiple objects, many times higher than the average rate of label noise.

Taking label-noise flagging a step further, in some cases, we can utilize the neural activation maps of core robust neural features for these classes to automatically refine the mislabeled high spuriosity images. Namely, we can crop the image based on the region that activates core features for the class, to focus on the actual object of interest and remove the spurious co-occurring one (details in Appendix H). Alternatively, segmentation models may be used to refine flagged samples.

## 6   Conclusion

In this work, we tackle the problem of spurious correlations in image classifiers by shifting attention from training algorithms to *data*. As datasets continue to grow, it can be challenging to answer even the simplest questions about what resides within them. We propose to leverage interpretability tools to harness machine perception towards scalably organizing data. Namely, we sort data within each class based on spuriosity, as measured by neural features in an interpretable network. With spuriosity rankings, we can easily retrieve natural examples and counterfactuals of relevant spurious cues from large datasets, enabling efficient measurement and mitigation of spurious feature reliance. We demonstrate the feasibility of our approach on a vast set of models on ImageNet – a far larger and more realistic setting than most prior spurious correlation benchmarks. Further, we observe highly similar biases across our diverse model set, suggesting that biases may be determined more so by what a model sees than how it is trained. We hope our work spurs further exploration into how to best understand and utilize the data we already have, towards more robust and interpretable models.

## 7   Acknowledgements

This project was supported in part by a grant from an NSF CAREER AWARD 1942230, ONR YIP award N00014-22-1-2271, ARO's Early Career Program Award 310902-00001, Meta grant 23010098, HR00112090132 (DARPA/RED), HR001119S0026 (DARPA/GARD), Army Grant No. W911NF2120076, the NSF award CCF2212458, an Amazon Research Award and an award from Capital One. MM also receives support from the ARCS foundation.

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

# Appendix

## A  Impact and Limitations

We hope our work will allow for better understanding of how spurious feature reliance affects deep models, leading to improved handling of algorithmic fairness and model robustness, which are both related to the spurious correlation problem. We open source all code, and modify a key framework of our method to vastly increase its usability for people with fewer computational or data resources. We also create a website for easier viewing of our analysis, making our entire annotation process completely transparent. We obtain verbal approval [*] from our institution's IRB for our study, and pay workers above minimum wage. Key limitations are that our method does not detect all spurious features (only a subset), and uses a single model to detect and measure them, though we argue that the features we detect are very relevant and likely relied upon by all models; we elaborate on this point in Section I.3.

## B  Characterizing Feature Sensitivities

Using visualizations from our robust neural feature annotation, we obtain soft segmentations for features in input space that activate a specific neural feature. Namely, the Neural Activation Map for a neural feature can serve as a soft segmentation for the feature. Using a Mechanical Turk human study, we validate that neural activation maps for images in the top $20^{th}$ percentile within its class for activation of the neural feature segment the same input pattern (see Appendix I.2.2).

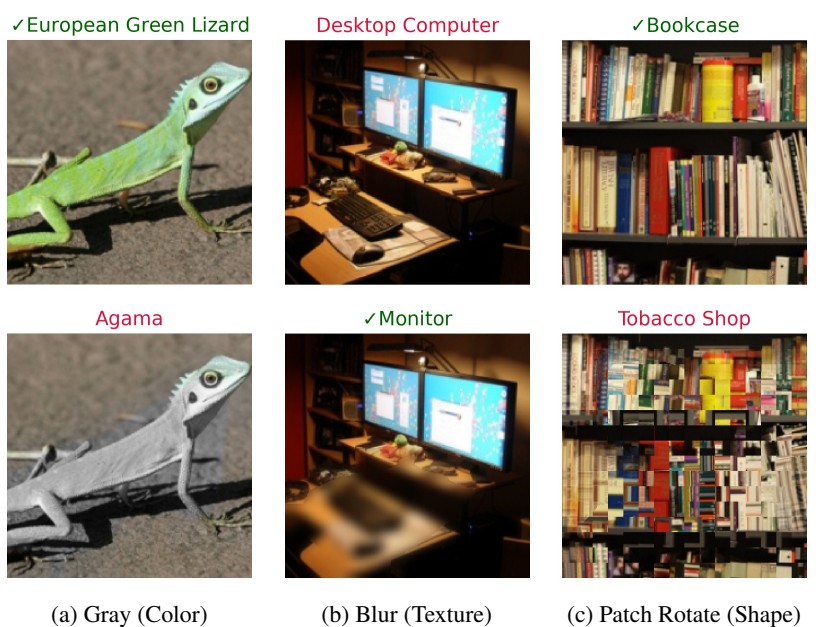

(a) Gray (Color)          (b) Blur (Texture)          (c) Patch Rotate (Shape)

Figure 10: Examples of features that are particularly sensitive to color, texture, and shape corruptions. Original images (**top**) and corrupted images (**bottom**) shown, along with model predictions. In one case, corrupting the spurious feature of *keyboard on desk* for class *monitor* actually improves prediction.

We then use targeted corruptions to gain insight on the information within an input region that models rely on most. The core premise of our argument is that degradation in model performance due to corruption of some information can be a proxy of the importance of the corrupted information. Indeed, similar arguments were made in [46, 29], where adding Gaussian noise to specific regions was employed to assess core/spurious or foreground/background sensitivity respectively. However, in our

[*]We were informed that formal approval is not needed because we do not obtain information *about* our crowd worker, and thus our study does not constitute human-subject research.

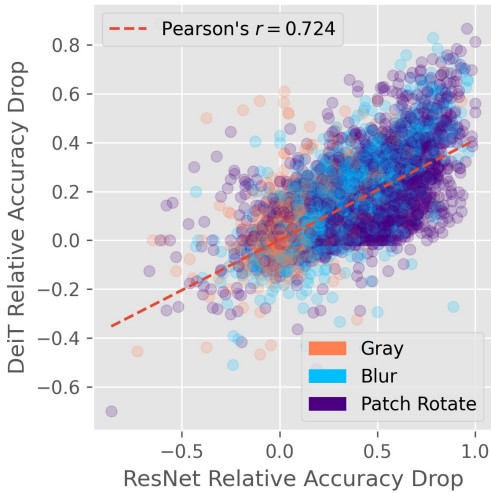

Figure 11: Sensitivities of each class-feature pair to specific corruptions for ResNet-50 and DeiT-Small. The two distinct models rely on features in the *same* way.

analysis, we use carefully chosen corruptions so to remove one aspect of primitive information (i.e. color, shape, or texture) without disrupting the others. Namely, our corruptions are graying out image regions, blurring out image regions, or patching an image region and randomly rotating each patch by a multiple of 90 degrees. Graying out an image removes color, blurring out destroys local (i.e. textural) details, and random patch rotation breaks longer edges, hence disrupting shape information.

We apply these targeted corruptions to segmented feature regions for all 5000 annotated class-feature pairs. Specifically, we focus on the 65 images per class where a feature is activated most highly (i.e. roughly top $5^{th}$ percentile), since the feature is most prominent in these images. Also, for spurious features, we include a filtering step to avoid corrupting core regions. We do this by removing any overlap of the spurious feature segmentation with the consolidated core mask [46], which is a pixel-wise maximum of soft segmentations for any relevant core features. We track model accuracy on samples with and without each corruption applied, using change in accuracy as a measure of sensitivity.

Figure 10 shows examples of features that are the most sensitive (i.e. highest change in accuracy) to each of the three corruption types, visualizing the corruption and its affect on model prediction. In the left panel, we see that the color information in the core feature of *lizard body* is crucial for successful prediction of the class *European Green Lizard* (matching intuition). In the middle panel, we see a case where the spurious feature of *keyboard on desk* hurts classification. Namely, blurring out the keyboard and desk leads to more accurate prediction of the *desktop computer*. We note that while the shape and color information is retained, it appears that the local details in the texture of the keyboard and desk contribute more to confusing the model and leading to misclassification. Lastly, for the spurious feature of *books* on a *bookcase*, we observe that the shape of the books is crucial, as rotating patches leads to an incorrect prediction of *tobacco shop*.

We conduct this analysis using two pretrained models of roughly equal size: ResNet50 and small DeiT. Figure 11 shows the relative accuracy drops incurred by performing each of the three corruptions for all 5000 annotated class-feature pairs on both models. We find that the two models have strongly correlated sensitivities ($r = 0.724$). The least correlated sensitivities appear to be for the patch-rotate corruption, where the transformer has nearly zero accuracy drop for many cases where the ResNet experiences greater accuracy drop. This is most likely due to the unique robustness of vision transformers to various patch transformations like permutation, rotation, and ablation [34, 36]. Despite this, the strong correlation between feature sensitivities for two diverse models suggests that model behavior is determined much more by the data it operates over than the specific decisions made during training. In other words, the features that models rely on and the ways in which they rely on them may be far more a function of data than a function of the model. If this conjecture holds, it would warrant greater inspection of the role of data in various open problems in modern deep learning (robust generalization, efficient learning, etc).

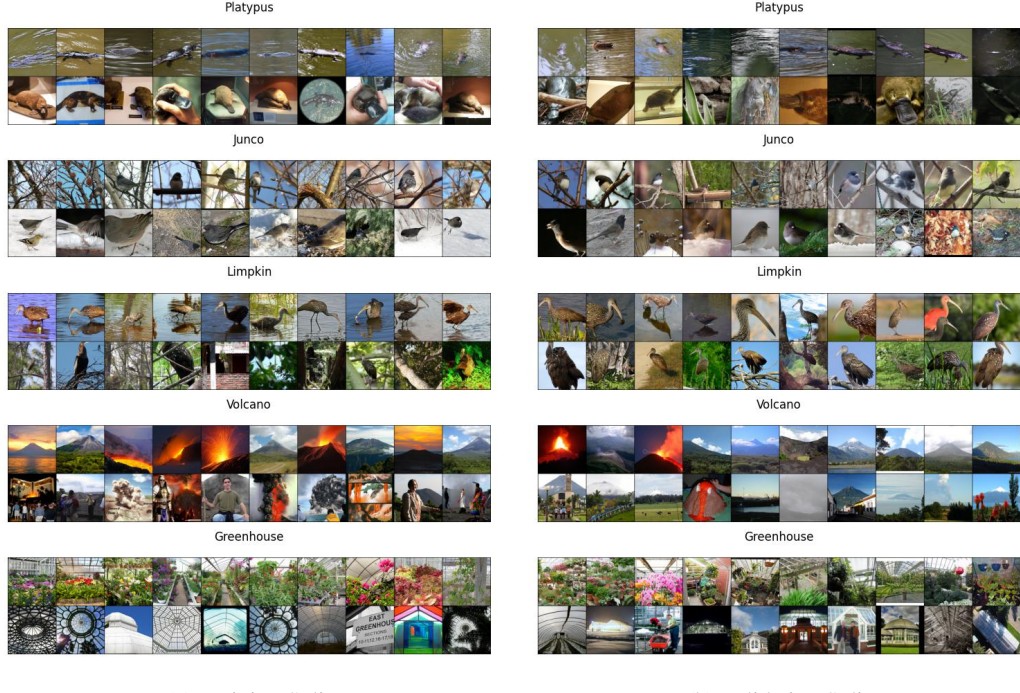

| (a) Training Split | (b) Validation Split |

Figure 12: Images with highest and lowest spuriosity for five randomly selected classes (top/bottom row in each chunk corresponds to images with highest/lowest spuriosity respectively). The spurious features depicted are *lake water, branches, water, sky with clouds or lava*, and *plants*. The spurious features are easier to see in the training split than in the validation split, because there are roughly 25× more samples to choose from. Nonetheless, the validation images still seem to be organized so that the top-10 images contain the spurious feature much more than the bottom-10, indicating that our spuriosity ranks generalize to the validation split.

## C  More Examples of Spuriosity Rankings

To further validate our spuriosity rankings, we now present more examples of images ranked by spuriosity. While this validation is qualitative, we argue that it is sufficient since we are attempting to proxy a fundamentally qualitative notion. Namely, in figure 12, we present images organized by spuriosity ranking for five randomly selected classes. Here, we show images both ranked in the training and validation split to demonstrate that even though trends are clearer when ranking training images (because there are $\sim 25\times$ more of them to choose from), the underlying semantic concepts by which images are ranked still generalize to the validation split.

## D  Full Pretrained Model Evaluation Results

We now provide greater detail for the pretrained model evaluation presented in Section 4.2. We evaluate 89 models spanning diverse architectures and training procedures. Namely, the models we consider fall into the following categories, with category nicknames in parenthesis: supervised vision transformers (Sup ViTs), self-supervised vision transformers (SS ViTS), CLIP vision transformers (CLIP ViTs, both Zero-Shot and Finetuned), CLIP ResNets (CLIP RNs), self- or semi-supervised ResNets (SS ResNets), supervised ResNets (Sup ResNets), mixer and ResMLP models (MLP-Based), various other convolutional networks (Other CNNs), and adversarially trained ResNets (Robust ResNets). The vast of majority of pretrained weights are obtained from the timm library [51], while others come directly from their original respective repositories [7, 9, 37, 43]. Figure 13 shows effective robustness for each model, as well as the average effective robustness per category.

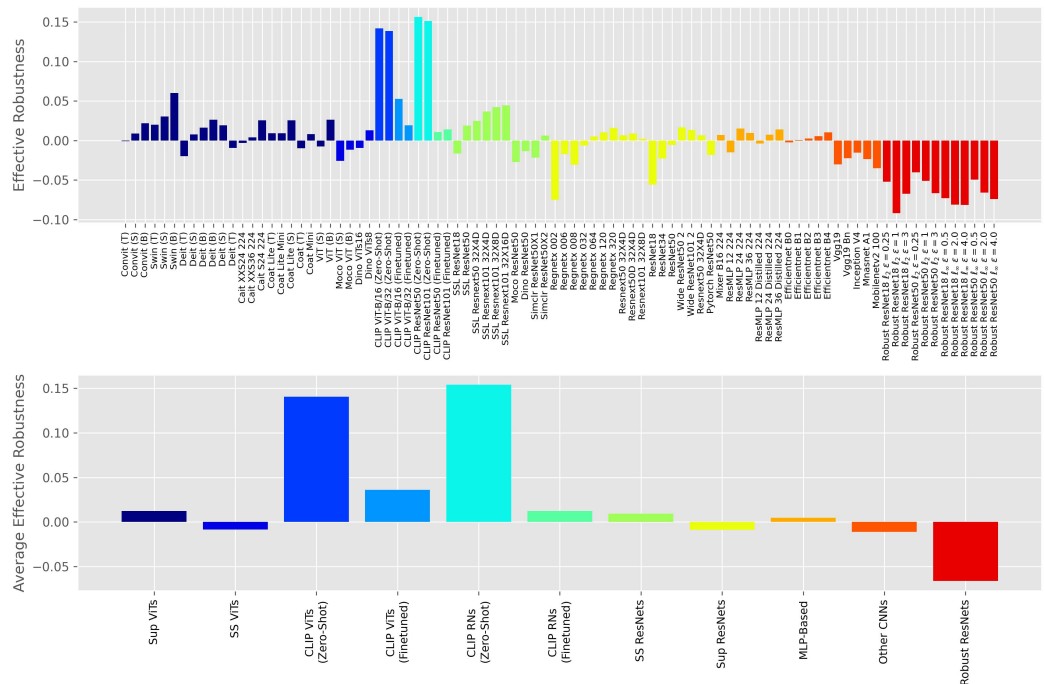

Figure 13: Effective robustness for all 89 models considered (**top**), along with effective robustness averaged over model category (**bottom**).

We measure effective robustness by first fitting a line to predict accuracy on the 10 validation images per class with the *least* spuriosity as a function of accuracy on the 10 validation images per class with the *most* spuriosity. Let $\text{acc}_{\text{top}}$ and $\text{acc}_{\text{bot}}$ denote accuracy on images with the most and least spuriosity respectively (i.e. top and bottom with respect to spuriosity rankings). We refer to the line of best fit as $\beta : \mathbb{R} \to \mathbb{R}$. Effective robustness for a model with accuracies on top and bottom spuriosity ranked images $\text{acc}_{\text{top}}, \text{acc}_{\text{bot}}$ respectively is simply $\text{acc}_{\text{bot}} - \beta(\text{acc}_{\text{top}})$. In other words, we measure how high above the line of best fit a model's marker falls in figure 6 (left).

We note that this differs slightly from the originally presented notion of effective robustness in [27], where a log-linear fit is performed. Moreover, instead of predicting out-of-distribution accuracy based on in-distribution accuracy, we compare accuracy on two carefully chosen subsets of in-distribution data. We argue that we capture subsets where subpopulation shift has occurred, specifically with respect to common spurious cues for each class.

Our findings show that some models are more effectively robust than others. Namely, zero-shot CLIP models have much higher accuracy on images with lowest spuriosity than their performance on the images with highest spuriosity would predict. After finetuning a linear head using ImageNet on top of the fixed CLIP image encoder, we see effective robustness drops significantly. We note that both these results were observed using independent distributional robustness measures in [53]. At the other extreme, adversarially trained ResNets have the lowest effective robustness, a result carefully studied in [28].

Aside from these two extremes, most trends are muted. We believe the stronger signal exists in per-class spurious gaps, as shown in figure 6 (right), suggesting that while models generally behave similarly on the same inputs, their behavior (i.e. with respect to spurious feature dependence) varies dramatically across inputs. Indeed, we find that per-class spurious gaps correlate strongly between pairs of models in our evaluation suite. Figure 14 visualizes these correlations, with an average correlation of $r = 0.69$ being observed. We thus recommend closer attention to be paid to the individual data classes and potential spurious features *per class* a model is to be deployed over.

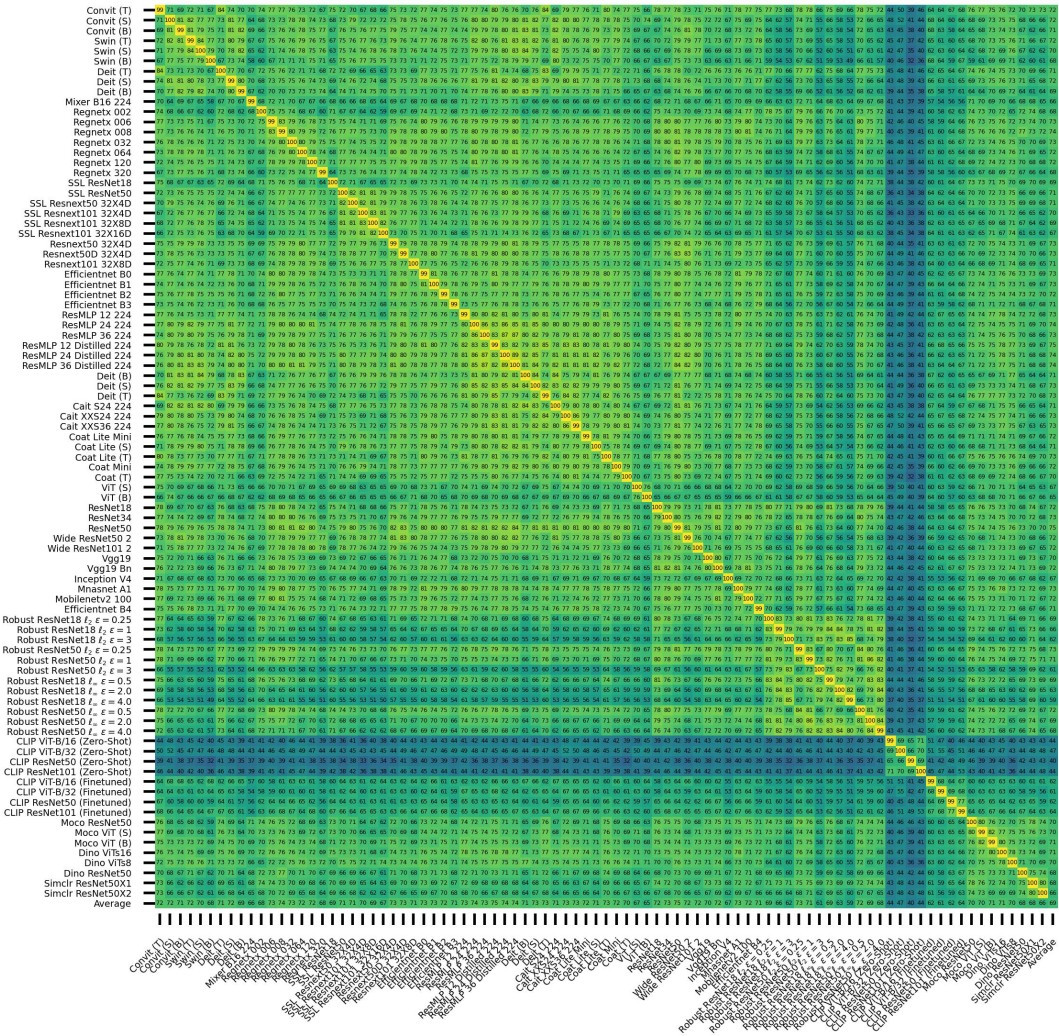

Figure 14: Correlation of per-class spurious gaps for all pairs of models. Numbers shown are Pearson's $r$ correlation coefficients times $100$. On average, a pair of diverse models have strongly correlated class-wise spurious gaps, with $r = 0.69$. Notably, zero-shot CLIP models have the worst correlation with other models, suggesting that their perception is fundamentally different.

## E  Details on Feature Discovery Beyond ImageNet

### E.1  General Training Details

Recall that we introduce a lightweight extension of spurious feature/concept discovery without requiring adversarial training by simply fitting a linear layer on new data over a fixed adversarially trained feature encoder (Section 3.2). Specifically, in all cases, we use a ResNet50 feature encoder (i.e. all layers except for the final linear classification head) adversarially trained for ImageNet classification with projected gradient descent using an $\ell_2$ norm attack of budget $\epsilon = 3.0$, downloaded from [43]. Note that this feature encoder comes from the same network used to discover core and spurious features in ImageNet. However, we train new linear heads over fixed features for four tasks over three datasets: Waterbirds [42], Celeb-A [25], and UTKFace [60]. In all cases, we use an Adam optimizer with learning rate of $0.1$ and weight decay of $0.003$. We train for 20 epochs. While the model we use may not be as accurate as a model trained from scratch, they still achieve relatively high accuracies given the ease of their training. Notably, for Celeb-A hair task (2-way), UTKFace gender task (2-way), and UTKFace Race task (5-way), our linear layer fitting on robust neural features yields

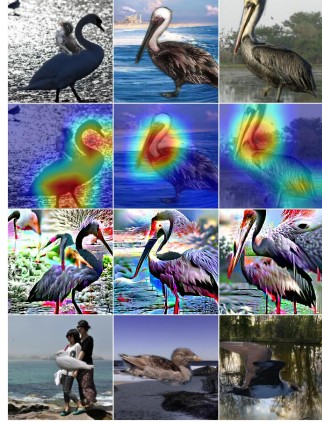 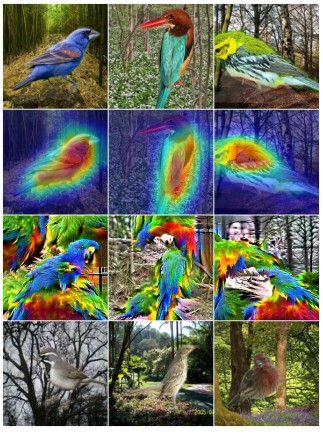

(a) Waterbird core feature. Focus is on elongated and curved necks.

(b) Landbird core feature. Focus is on feathers with bright colors.

Figure 15: Core features discovered for the Waterbirds waterbird vs. landbird classification task. We identify features that occur in specific subpopulations of the broader classes. Low activating images reveal subpopulations where the feature is absent. The rows show (from top to bottom): highly activating images, corresponding heatmaps and feature attacks, and lowly activating images.

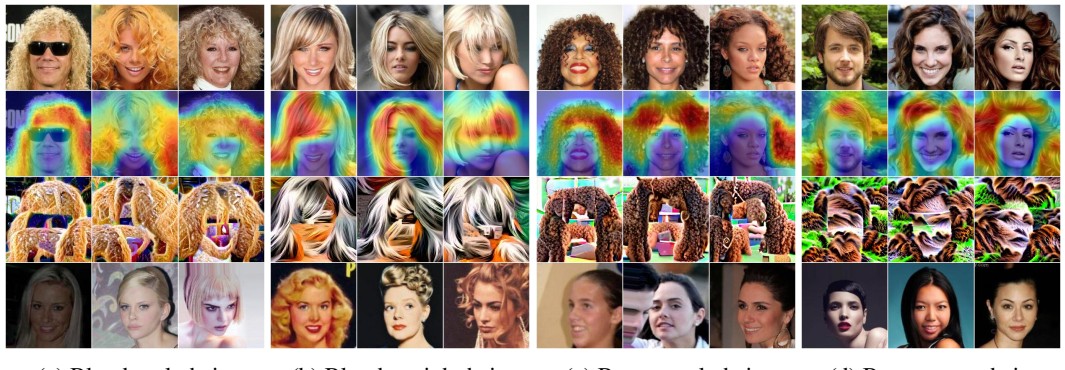

(a) Blond curly hair.      (b) Blond straight hair.      (c) Brown curly hair.      (d) Brown wavy hair.

Figure 16: Core features discovered for the Celeb-A blond vs. brown hair classification task. We identify features corresponding to different types of hair within the same broader class of hair color. The rows show (from top to bottom): highly activating images, corresponding heatmaps and feature attacks, and lowly activating images. The captions reflect the focus of the robust neural feature.

accuracies of $95.3\%, 86.5\%$, and $66.7\%$ respectively. Acccuracy on the Waterbirds validation set is lower ($77.6\%$), though this is typical for Waterbirds, as the validation shift features a significant subpopulation shift (by design, due to breaking the background correlation). More importantly, in all cases, important neural features are observed to correspond to data patterns (i.e. human concepts) salient to the task at hand, including spurious ones.

## E.2    Additional Discovered Features for Waterbirds and Celeb-A

We now present additional features identified via our lightweight extension of the feature discovery framework of [46]. Specifically, we show core features, which are relied upon more frequently than spurious features, which we show in Figure 4. We observe robust neural features to respond to specific patterns shared among subpopulations within the broader classes. Surprisingly, these robust neural features were not at all trained on images from the downstream tasks. Yet, the patterns they respond to are semantically meaningful for the downstream task. We attribute this to the diversity in ImageNet, which leads to learning many general visual patterns that are manifested in more specific contexts for a vast number of downstream tasks.

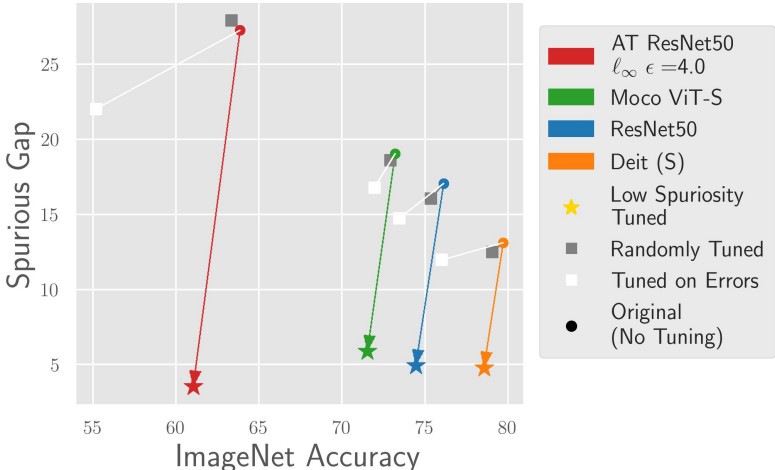

Figure 17: In addition to tuning on low-spuriosity images and on random samples, we include results from tuning on misclassified samples in white. Error tuning minimally closes spurious gaps, and actually leads to substantial drops in overall validation accuracy.

### E.3 UTKFace Experiments

UTKFace [60] is an attributed dataset originally intended for the task of synthetically aging a face image using conditional generative models. The dataset consists of over $20,000$ images with age, ethnicity, and gender annotations. We consider the task of gender and age classification. We use $90\%$ of the data for training, and hold out the remaining $10\%$. We inspect the 15 most important features for each class per task. For all classes and tasks, we are able to identify at least on spurious feature with our method. Notably, these tasks were not designed to contain spurious features, like the other two benchmarks analyzed. This goes to show that spurious correlations may arise in any setting, and our method can reveal such spurious features with minimal computation. Having discovered robust neural features that detect spurious concepts, one can sort data by spuriosity to measure and mitigate the resultant bias.

## F    Alternate Baseline: Error Tuning Does Not Close Spurious Gaps

A common approach within recent spurious correlation literature is to automatically select samples to conduct additional training on by simply inspecting whether they were classified correctly originally. For example, Liu et al. [24] propose to 'just train twice', upweighting error samples in the second round of training; Nam et al. [32] 'learn from failure' by training one classifier that is intentionally biased, and using the hard (i.e. misclassified) samples from the biased classifier to obtain an unbiased classifier; Zhang et al. [59] use error samples as hard positives in an additional round of contrastive training. Inspired by these approaches, we include an additional baseline to the experiment from section 4.3 where we tune existing classification heads on samples originally misclassified. Specifically, we train on misclassified samples for all classes. All three methods train on 100 samples per class. When the number of errors in a class is less than 100, we add randomly selected correctly classified class instances. When the number of errors is more than 100, we randomly select 100 of them. Results shown are averaged over three trials.

We observe that while error tuning closes spurious gaps more than tuning on random samples, the reduction in spurious gaps is far less than that from low spuriosity tuning. More importantly, overall validation accuracy drops substantial when tuning on errors. We conjecture this occurs because errors can be caused by many things aside from the absence of spurious cues, such as label noise. Thus, tuning on misclassified samples could lead to overfitting on unreliable data. In contrast, low spuriosity images are designed to only differ from typical samples in that they lack particular cues that have already been deemed spurious by a human. Arguably, low spuriosity images can at times offer more reliable learning signal, as they do not contain distracting shortcuts (e.g. the low spuriosity lighters in figure 1 are far easier to see than the high spuriosity ones). Also, while the error-centric

approaches were shown to be effective on existing spurious correlation benchmarks, we hypothesize that the biases in those benchmarks are overly simplistic: they typically only include one spurious correlation and at most a handful of classes. Therefore, these methods may struggle in more realistic settings, as demonstrated by their ineffectiveness on this ImageNet scale experiment.

Thus, we claim that low spuriosity tuning offers unique advantages compared to prior work. While most existing methods focus on altering the training algorithm, our method acknowledges that data may have a larger (and often overlooked) role in determining the biases a model absorbs, and focuses on finding the right data to tune. The experiments in this section show that compared to another line of data-centric approaches, our low spuriosity tuning may be more effective, in both retaining overall accuracy and reducing biases.

## G   On Automating Spuriosity Rankings

We believe that human involvement is a strength of our framework, as it increases transparency. Namely, the human in the loop is given a concise inside look on the cues a model trained on the given data is likely to rely upon. Moreover, the human is given agency to decide which cues model performance should be invariant to. Also, the level of human involvement is relatively low. Given the complexity of the task of sorting thousands of images within a class, only requiring a human to inspect a handful of images is relatively efficient. With an appropriate UI, we can confirm that this takes no longer than about a 30-40 seconds per class.

However, in cases where minimizing cost is preferred, Spuriosity Rankings can be automated. One way to do so is to automate the annotation of neural features as core or spurious. Specifically, one could automatically segment the class object with open-vocabulary segmentation models [21], and then compute the amount of saliency placed on the object by the neural activation map. If most of the salient pixels for a neural feature lie outside of the image region containing the class object, one could automatically flag such a feature as spurious.

Another way to automate Spuriosity Rankings is to leverage vision-language models (VLMs) like CLIP [37]. With VLMs, we can compute the similarity of an image to any concept, encoded directly from text. After sorting images by similarity to spurious concepts, we can inspect the Spurious gap to measure bias, and train on low spuriosity images to mitigate bias. Thus, one must simply enumerate (in the form of text) potential relevant spurious cues; since encoding the cue and sorting along it is free, one can discard the cues which do not result in an accuracy gap between the most and least similar images, as the model of interest apparently would not be biased to the presence of that cue. We note that while VLM training is expensive, a pre-trained model would suffice, and moreover, recent work shows that VLM abilities can be extended to arbitrary vision-only models cheaply as well [31].

Note that the underlying mechanism of Spuriosity Rankings is preserved in both of these automateed variants: we utilize the representation space of an interpretable model to quantify the presence of relevant cues, and rank images by spuriosity (proxied by similarity/activation of concept directions in representation space) to enable interpretation of spurious cues in context and measure model bias caused by them.

## H   Details on Core-Cropping

To perform core-cropping, we first obtain a soft segmentation of core regions, done by averaging the neural activation maps for core neural features. Next, we threshold the averaged mask to only retain highly activated pixels. We use a threshold of $0.9$. Then, we obtain a bounding box to localize a rectangular region encompassing the pixels that most highly activate the core region. Since we use a high threshold, we expand our bounding box by $20\%$ along both height and width dimensions. Finally, we convert our rectangular region to a square one by simply extending the shorter side to match the length of the larger side. We note that this method is only as effective as the core soft segmentation. Note that core neural activation maps are generally reliable for images that reasonably activate the core features. Namely, we validate the quality of these segmentations for images in the top $20^{th}$ percentile of activation for a class (see Appendix I.2.2). However, for the remaining $80\%$ of images, these soft segmentations may not be perfectly reliable. Nonetheless, as segmentation models improve, core-cropping may also improve, as we can simply replace core soft segmentations obtained

via neural activation maps with actual segmentations, conditioned on the class object. Thus, flagging potentially mislabeled samples

# I  Robust Neural Feature Annotation Details

Our work uses the feature discovery framework introduced in [46], also known as Salient ImageNet. We now i. introduce a taxonomy of relevant terms, ii. review the framework, iii. discuss its merits and potential limitations, and iv. detail our expansion/modifications to it.

## I.1  Taxonomy of Relevant Terms

Below we enumerate and explain some terms we refer to throughout the paper.

- **Core** feature or cue: a visual pattern that is essential to the class object; i.e., the pattern is a part of the class object, like fur for an otter.

- **Spurious** feature or cue: a visual pattern that is not essential to the class object; i.e., the presence of the spurious cue is not required for the image to belong to the class, and the spurious cue can exist in images from other classes.

- **Neural Feature**: node in the penultimate layer of a classifier parameterized by a deep neural network. Importantly, each logit of the classifier is a linear function of neural features.

- **Robust neural feature**: neural feature from an adversarially trained network. We use an $\ell_2$ PGD-trained [26] network with $\epsilon = 3$.

- **Neural activation map**: an extension of a class activation map [61] for neural features, used to highlight the region of an image responsible for activating a neural feature.

- **Feature attack**: a visualization technique to amplify the visual cues present in an image that activate a neural feature.

## I.2  Salient ImageNet Framework

The steps of the Salient ImageNet Framework are:

1. Adversarially train a model for your classification task.

2. Identify important neural features per-class based on feature importance, which is simply the product of the average feature activation and the weight of linear head connecting feature to class logit. Feature importance explicitly computes the average contribution of a feature to a class logit for samples in the class.

3. Select the top-$k$ (i.e. $k = 5$) activating instances of a class for a selected feature and additionally generate heatmaps (via overlaying the Neural Activation Map on the original image) and feature attacks (via adversarially attacking the original image using gradients from the adversarially robust model so to maximize activation on the feature).

4. Have humans inspect the $3 \times 5 = 15$ visualizations to determine if the focus of the neural feature is on the main object (i.e. pertaining to the class and hence *core*) or on the background or a separate object (hence making the feature *spurious*).

We elaborate on the feature discovery and annotation procedure in subsection I.2.1, including an example of the MTurk form. In [46], the annotated robust neural features were primarily used to softly segment image regions containing core and spurious features at scale. Thus, in addition to the Mechanical Turk study performed to annotate neural features, a second study was conducted to confirm that neural activation maps for annotated features softly segment the same input features across the top $5^{th}$ percentile of samples within the relevant class. With this confirmation, they computed neural activation maps for all of these images and used these as feature segmentation maps. To assess reliance on core and spurious features, they would inspect drop in accuracy due to adding small amounts of Gaussian noise to these regions. We repeat a modified and expanded version of this second study, so to validate a larger number of feature segmentations. Details on this human study are in subsection I.2.2.

### I.2.1 Mechanical Turk study for discovering spurious features

The design for the Mechanical Turk study is shown in Figure 18. The left panel visualizing the neuron is shown in Figure 18a. The right panel describing the object class is shown in Figure 18b. The questionnaire is shown in Figure 18c. We ask the workers to determine whether they think the visual attribute (given on the left) is a part of the main object (given on the right), some separate object or the background of the main object. We also ask the workers to provide reasons for their answers and rate their confidence on a likert scale from 1 to 5. The visualizations for which majority of workers selected either background or separate object as the answer were deemed to be spurious. Workers were paid $0.1 per HIT, with an average salary of $8 per hour. In total, we had 137 unique workers, each completing 140.15 tasks (on average).

We conduct this study for the five neural features with highest contribution to each of the 768 ImageNet classes not analyzed in [46], resulting in 3840 newly annotated class-feature dependencies, including 468 spurious ones ($3.3\times$ and $2.9\times$ increases respectively).

### I.2.2 Mechanical Turk study for validating heatmaps

We now detail the second MTurk study we carry out to validate the feature soft segmentations generated by Neural Activation Maps of annotated robust neural features. For each annotated class-feature dependency, we first obtain the training images from the class who's activation on the feature is within the top $20^{th}$ percentile for the class. This results in a set of 260 (20% of 1300 training images per class) images per class-feature pair, as opposed to only 65 in [46]. From this set of 260 images, we selected 5 images with the lowest activations on the neural feature and randomly selected 10 images from the remaining set (excluding the already selected images). We show the workers three panels: images and heatmaps with (i) the highest 5 activations, (ii) the next 5 highest activations, and (ii) the lowest 5 activations. For each panel, workers were asked to determine if the highlighted attribute looked different from at least 3 other heatmaps in the *same panel*. Next, they were asked to determine if the heatmaps in the 3 *different panels* focused on the same visual attribute, different attributes or if the visualization in any of the panels was unclear. Thus, we validate that both within each panel and across panels, the focus of the neural feature is consistent.

The design for the Mechanical Turk study is shown in Figure 19. The three panels showing heatmaps for different images from a class are shown in Figure 19a. The questionnaire is shown in Figure 19b. For each heatmap, workers were asked to determine if the highlighted attribute looked different from at least 3 other heatmaps in the same panel. We also ask the workers to determine whether they think the focus of the heatmap is on the same object (in the three panels), different objects or whether they think the visualization in any of the panels is unclear. Same as in the previous study (Section I.2.1), we ask the workers to provide reasons for their answers and rate their confidence on a likert scale from 1 to 5. The visualizations for which at least 4 workers selected same as the answer and for which at least 4 workers did not select "different" as the answer for all 15 heatmaps were deemed to be validated i.e for this subset of 260 images, we assume that the robust neural feature's focus is on the same visual attribute. Showing three panels as opposed to two is novel to our work and increases the standard for validation.

For all (class, feature) pairs ($1000 \times 5 = 5000$ total), we obtained answers from 5 workers each. We successfully validate 4763 pairs (i.e. $95.26\%$), demonstrating the effectiveness of using neural feature supervision to automatically (softly) segment an input pattern in a large number of images. We utilize these feature segmentations in Appendix B.

### I.3 On Feature Discovery Using Only One Model

The key idea of our framework (expanded from [46]) was to use neural features of an adversarially trained network as automatic 'concept' detectors, and further to use their activation maps as soft segmentations. One may argue then that the features discovered are biased to the underlying model used. However, we argue that the primary patterns that models rely on are generally the same. Note that the existence of patterns predictive of a class is determined by the data distribution and not the model. While models may learn to rely on different patterns and may also vary the degree to which they rely on the same patterns, we believe that the most predictive patterns will be detected and relied upon by nearly all models. In our experiments, we find that the class-wise spurious gaps are highly correlated for any two pairs of models, with average correlation of $r^2 = 0.69$ (see figure 14). We also observe

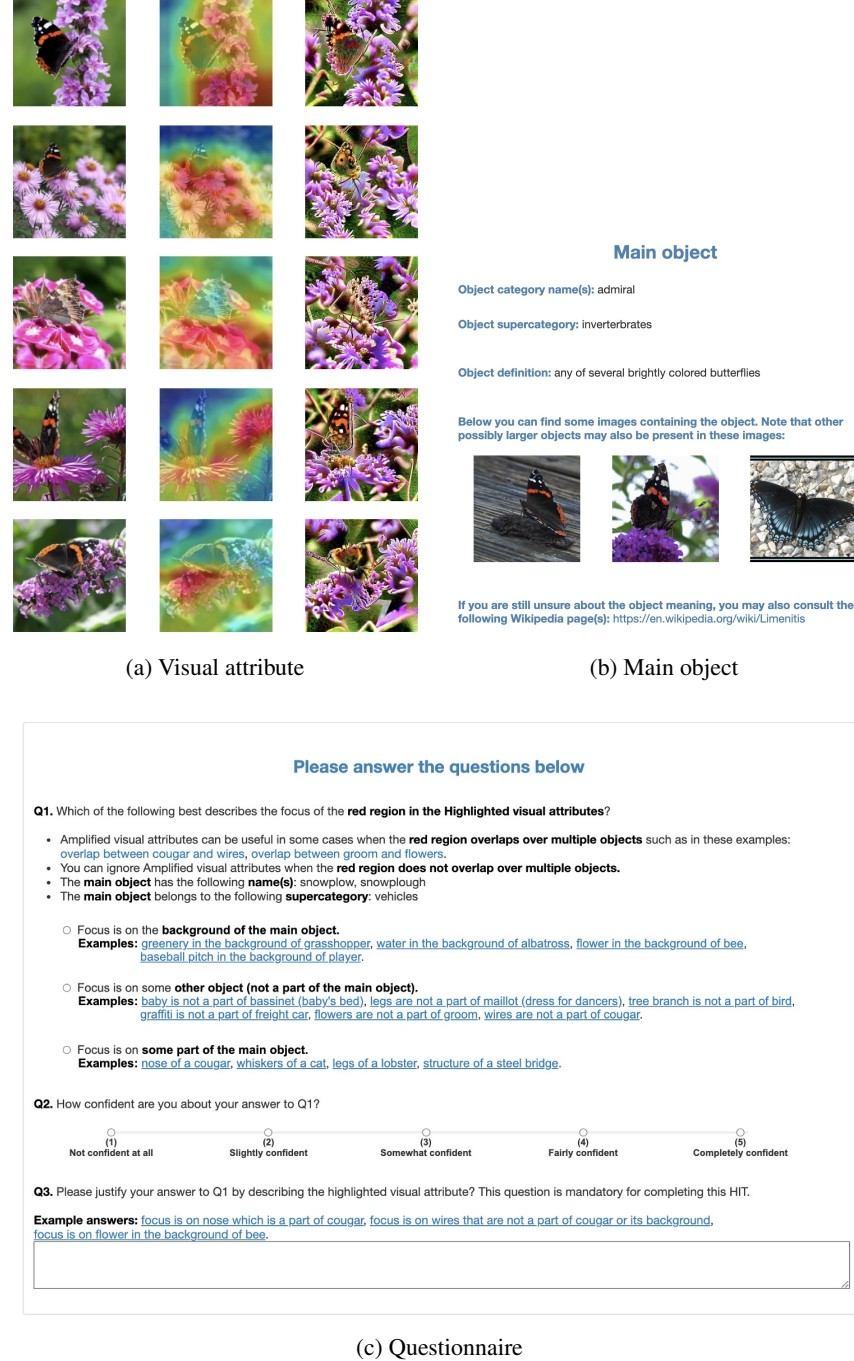

(a) Visual attribute

(b) Main object

(c) Questionnaire

Figure 18: Mechanical Turk study for discovering spurious features. This figure is from Singla and Feizi [46] and included here for completeness.

that the accuracy drops experienced due to targeted corruption of input regions where certain features reside for a ResNet and a vision transformer are highly correlated too ($r^2 = 0.724$) (see figure 11). These results suggest that models are sensitive to the same data patterns and even rely on the different pieces of information (i.e. color, shape, texture) captured in the data patterns in the same ways.

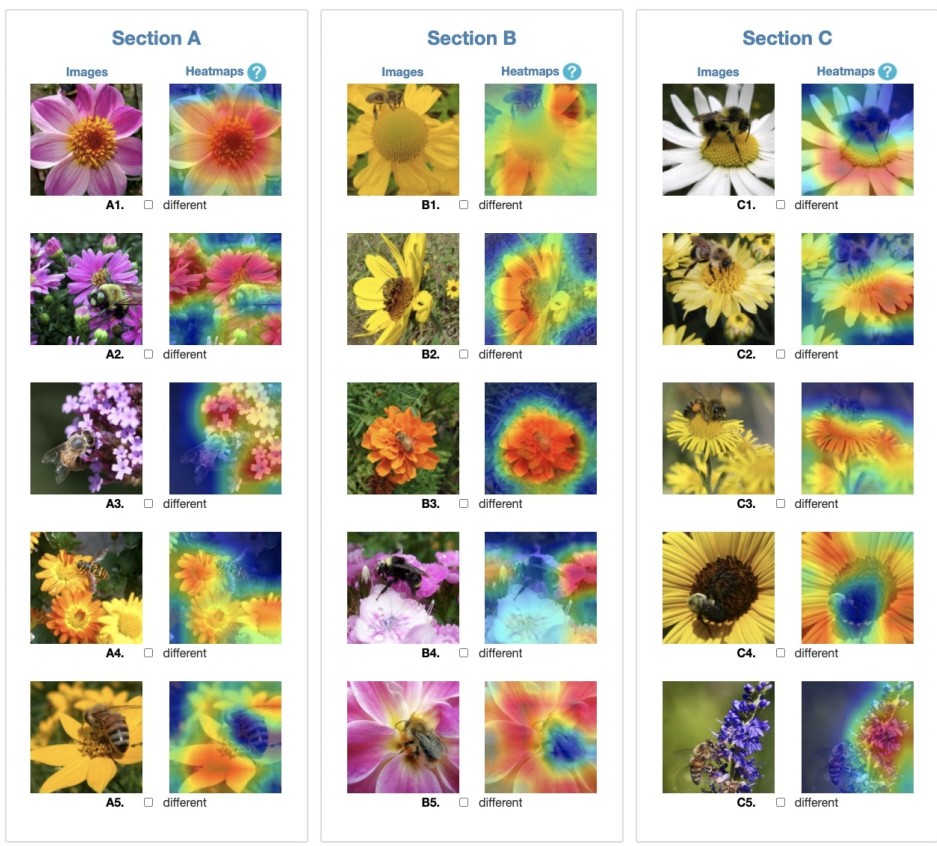

(a) Heatmaps highlighting the visual attributes

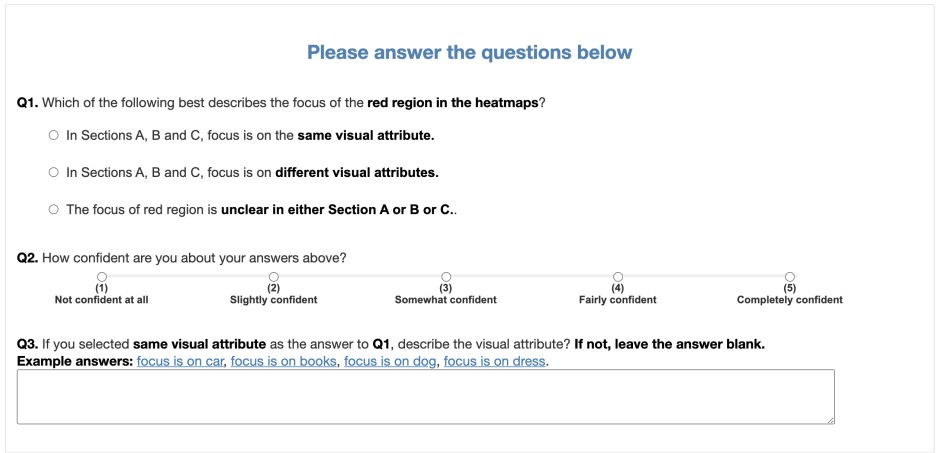

(b) Questionnaire

Figure 19: Mechanical Turk study for validating heatmaps

To be perfectly clear, we do not claim to have captured *all* spurious (or core) dependencies, mainly because we only inspect the 5 most important robust neural features per class. We acknowledge that another reason for certain dependencies to be overlooked is because we use a single model to detect and measure the presence of concepts. However, this only occurs for patterns that other models are sensitive to but the adversarially trained model is not. The main patterns that fit this description are adversarial ones, which contain no semantic meaning and thus arguably *should* be overlooked for

our purposes. In other work, adversarialy trained models were found to have increased reliance on spurious features [28]; this result was also corroborated in our spurious gap analysis. Thus, if anything, it seems like using an adversarially trained model may overlook core features, favoring spurious ones. Seeing as the goal of our analysis is to detect and assess spurious feature dependencies, we believe using robust neural features from an adversarially trained model is warranted, especially given the enhanced interpretability of this model, which allows for reliable heatmaps and feature attacks.

## I.4   Our Contributions

We now detail the novel contributions we make to the Salient ImageNet framework and analysis from [46]. The most crucial contribution is the notion that to interpret and improve spurious correlation robustness, one can seek to simply reorganize the data they already have, as opposed to collecting more samples or additional instance-wise supervision (i.e. segmentations, manual annotations beyond class-label) that would require fundamental changes the training strategy. Using spuriosity rankings, we gain new and surprising insight into the nuanced ways that models rely on spurious features. Namely, we find that contrary to common belief, models can perform worse when spurious features are present than when they are absent (i.e. because of *spurious feature collision*, as some spurious features are depended on for multiple classes). Further, our spuriosity rankings naturally lends itself to a simple, effective, and efficient manner to both measure the biases caused by spurious feature reliance, and mitigate theses biases for any model. Finally, we make a crucial adjustment to the feature discovery framework of [46] that removes its most costly and at times prohibitive step (adversarial training). Observing that the framework can be applied by simply fitting a linear layer atop a frozen adversarially robust feature encoder vastly increases the usability of this framework, cutting the time and data required to perform feature discovery by huge margins.

We now take a closer look at our method for measuring bias caused by spurious feature reliance compared to the one proposed in [46], as we believe our method corrects crucial shortcomings of the prior work. Recall that the prior evaluation framework consisted of corrupting core or spurious regions by adding Gaussian noise multiplied by soft segmentations of core/spurious regions obtained via Neural Activation Maps of annotated robust neural features. The drop in accuracy due to these regional corruptions was used to measure the importance of the region. We enumerate the shortcomings of this prior framework, and explain how our method evades these concerns, below.

- **Introduction of a synthetic distribution shift.** The previous framework required taking images out of distribution by synthetically corrupting core and spurious regions with Gaussian noise. Model behavior on these non-natural images may not truly reflect how the model would behave on the more realistic distribution shift caused by breaking spurious correlations.

- **Instability.** Adding noise introduced hyperparameters, like the amplitude and norm of the noise, which we observe the metric to be undesirably sensitive to. In our method, the only hyperparameter is the number of images we include in the sets of highest and lowest spuriosity images. We verify our results our stable with respect to this hyperparameter.

- **Inability to compare across diverse models.** Because models have varying robustness to Gaussian noise (e.g. some more recent models, especially vision transformers which utilize heavier regularization to compensate for lacking inductive biases of convolutional networks, use noising as a data augmentation during training), drop in accuracy due to Gaussian noise cannot be reasonably compared as a means to measure how models use the corrupted image region. Our metric utilizes accuracy, which is standard and established for model comparisons. Moreover, we can appeal to the notion of effective robustness [27] to further control for varying accuracies of the models we consider.

Other contributions that are less novel but still impactful include massively expanding the scale of the feature annotation analysis and creating a web interface to view the discovered dependencies. Specifically, we annotate over $4\times$ more classes, yielding interpretation of feature dependencies for all 1000 classes in ImageNet. Given that this benchmark is ubiquitous, we believe a deep dissection of what one model perceives as important patterns in ImageNet can be extremely informative in better understanding how machine perception differs from human perception. We also perform the MTurk study to demonstrate the generalizability of our feature annotations for the top $20^{th}$ percentile of images based on feature activation as opposed to the top $5^{th}$ percentile, leading to reliable feature

Class: 350 (Alpine Ibex), Feature: 389

This feature is *spurious* for class *Alpine Ibex*. It is the third most important feature for class *Alpine Ibex*.

Below, we display the five Alpine Ibex images that most highly activate the feature, along with the Neural Activation Map (heatmap) and Feature Attack for each image. These are the same as the images shown to crowdworkers to annotate this class-feature pair.

We additionally provide the full responses of the five crowdworkers that annotated this pair, as well as links to other related pairs (i.e. same class or same feature).

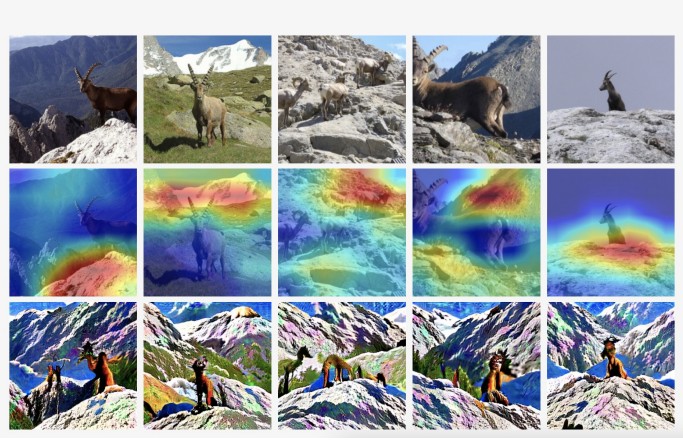

**MTURK WORKER ANNOTATIONS**

We now show the responses from five crowdworkers used to annotate this class-feature pair as *core* or *spurious*. Workers first denote if the focus of the image is on the main object (*core*), or a separate or background object (*spurious*). They also mark their confidence on a scale of 1 to 5, and provide written explanations for their annotation.

| Main Focus | Confidence | Explanation |
|---|---|---|
| Main Object | 4 | Focus is on some part of the main object ibex, capra ibex leg and head are focus |
| Background | 3 | The main object is capra ibex but the focus is on background. |
| Background | 5 | Focus is on mountain in the background of capra ibex |
| Main Object | 3 | Red focus is on ibex. |
| Separate Object | 5 | Focus is on mountain which is not part of ibex |

Votes for core: 2, Votes for spurious: 3. Feature is thus annotated as *spurious* for the class.

**RELATED CLASS-FEATURE PAIRS**

IMPORTANT FEATURES FOR *ALPINE IBEX*

There are 3 features annotated as *core* and 2 features annotated as *spurious* for *Alpine Ibex*. Core features include 462, 1207, 1526. Spurious features include 389, 820.

CLASSES WHERE FEATURE 389 IS AMONG MOST IMPORTANT (I.E. TOP-5)

Feature 389 is *core* for the following classes: Mountain, Valley, Volcano.
Feature 389 is *spurious* for the following classes: Marmot, Alpine Ibex, Ski.

Figure 20: Example of a landing page for one class-feature dependency. We present all visualizations shown to MTurk workers to annotate the feature as cor or spurious, as well as the annotations and explanations given by the MTurk workers. Further, we provide links to pages for other features relevant to the class, and other classes that also rely on the feature.

soft segmentations for many more images. Lastly, we create a website, which, while simply a user-interface, will greatly increase ease of viewing and transparency into our analysis, as we show all the visualizations used to perform our core/spurious neural feature annotations. The website can be found at `https://salient-imagenet.cs.umd.edu`. We present screencaps of pages from the website in Figure 20 and Figure 21. As a web-interface, our work is now accessible to people who do not code or lack the resources to generate our visualizations on their own devices.

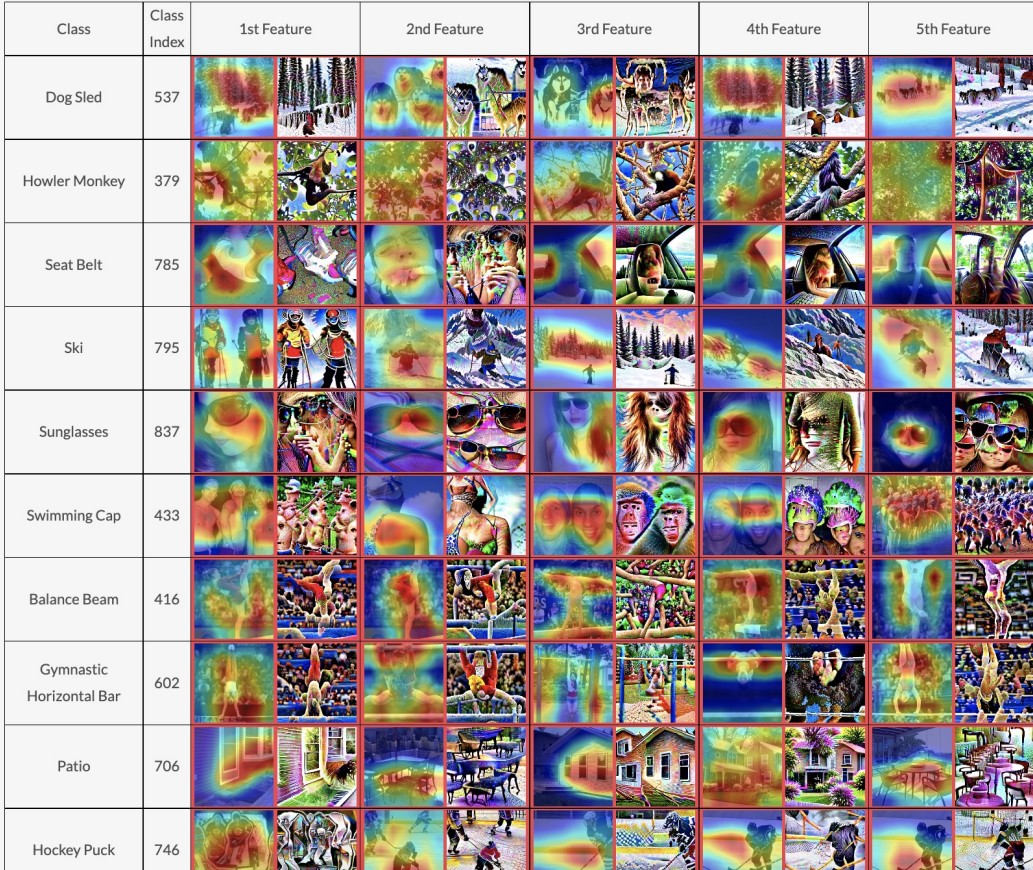

Figure 21: Screencap of a main table from our web-interface for accessing all 5000 class-feature dependencies. One can simply search for a class by its name or ImageNet class index to see the five features deemed most important for (i.e. contributing the most to the logit of) the class. Each image then links to an individual landing page designated for the class-feature pair (see Figure 20 for an example).

