# OpenReview forum: "Spuriosity Rankings: Sorting Data to Measure and Mitigate Biases"
_NeurIPS.cc/2023/Conference — NeurIPS 2023 spotlight_

### Official Review · Reviewer_y13p · 2023-07-03

**Soundness:** 3 good
**Presentation:** 2 fair
**Contribution:** 3 good
**Rating:** 6
**Confidence:** 4

**Summary:**

Shorcut learning has received increasing attention from the community recently. The paper proposes measuring and ranking data by "spuriosity," or the degree to which relevant spurious cues are present, as a way to detect and mitigate biases in deep models that arise from their tendency to rely on spurious correlations in the data. The authors use an interpretable model to identify neural features relevant to a class and then select the spurious ones based on limited human supervision. Ranking data by the activation of these spurious features yields many benefits, like revealing less biased subsets, quantifying model bias via "spurious gaps" in accuracy between high and low spuriosity data, and finetuning models on less spurious data to obtain robust performance. Analyzing spurious gaps for 89 models reveals that all underperform on less spurious data, indicating that bias stems more from what data models are trained on rather than how they are trained, and that spuriosity rankings provide an efficient and interpretable method to complement model-centric bias mitigation approaches.


**Strengths:**

1. This paper works on a timely and important problem. The paper is overall well written (except its dependence on a related work [42]) and I enjoy reading it.
2. It provides a scalable method to discover and rank data by how much spurious features they contain, i.e., spuriosity, which can help detect and mitigate biases in models.
3. This paper proposes to finetune models on low spuriosity data to improve performance on less biased instances while maintaining overall accuracy, resulting in more robust and more stable models.
4. Experiments results are insightful. The analysis of spurious gaps for many models indicates that biases stem more from the data than how models are trained, suggesting that data-centric approaches like this paper's can complement standard model-centric bias mitigation techniques.


**Weaknesses:**

1. One major concern is that this work is heavily based on a related work [42], which seems like from the same team of this submission. This makes the writing of this paper less friendly for readers not familiar with [42]. The authors are encouraged to revise the writing to make it more self-contained.
2. In line 176-177, the authors claim that “we are the first to uncover this racial bias in the Celeb-A benchmark”. Actually, this racial bias issue in Celeb-A has been well studied in various literature.


**Questions:**

None

**Limitations:**

Yes, the authors adequately addressed the limitations.

---

> ### Author Rebuttal · Authors · 2023-08-09
>
> We thank the reviewer for their time and comments. Also we appreciate the reviewer for saying the problem we tackle is ‘timely and important’, our ‘results are insightful’, and ‘the paper is well written’. We are very happy to hear you ‘enjoyed reading it’.
>
> **Clarity**
> We thank the reviewer for suggesting how to further improve our presentation, specifically w.r.t. explaining the mentioned prior work, from which we utilize their spurious feature discovery method. We note that we made multiple intentional efforts to make our work self-contained; we draw attention to these now. In the main text, we explain the method from prior work in several spots, each time providing more details: first at a high level in the introduction (L37-41), then in more depth in the related literature (L113-118), then in extensive detail in Section 3.1 (L132-148), where we also later explain our novel contributions atop the prior work (L155-161). Second, we design multiple visualizations to explain the framework, including images demonstrating specific methodologies from [42] (figures 2 and 3). Lastly, we note that we discuss the mentioned prior work in great detail in Appendix F, which includes a subsection (F.1) intended precisely to review the prior work so as to keep our paper self-contained.
>
> To further aide the extensive explanations we detail above, we will add the following to the revised draft:
> More explicit mentions to Appendix F to guide readers who desire more details.
> An additional appendix section that serves as a taxonomy for terms like ‘robust neural feature’, ‘feature attack’, ‘neural activation map’, etc.
> We hope these amendments will improve the presentation of the paper in the eyes of the reviewer, and if they do, we humbly ask that the reviewer increase their score for ‘presentation’.
>
> Lastly, we highlight that while we utilize a method from the prior work, we make several novel and impactful contributions atop it, detailed in L155-161 and at length in Appendix F.3.
>
> **Racial Bias in Celeb-A Hair Classification**
> We are unaware of any mentions of racial bias in the Celeb-A hair classification task in prior work. We conducted an additional review of literature, and were unable to find such work. While [1] mentions that Celeb-A likely has a bias toward images of white people, we note that this bias is different than the one we claim to discover. We refer to how “a model may spuriously associate brown skin with the brown hair class, 176 likely leading to failures for blond-haired brown-skinned people” (L175-176), and only discuss Celeb-A in the context of the hair classification task, a known benchmark in spurious correlation literature [2]. If the reviewer has a different citation in mind, we would be more than happy to include the citation and walk back the claim. Furthermore, we will change ‘Celeb-A benchmark’ to ‘Celeb-A hair classification benchmark’ to make this even clearer.
>
> [1] FairFace: Face Attribute Dataset for Balanced Race, Gender, and Age for Bias Measurement and Mitigation, Karkainen et al, WACV 2021
>
> [2] WILDS: A Benchmark of In-the-wild Distribution Shifts, Koh et al, ICML 2021

---

### Official Review · Reviewer_9RXj · 2023-07-06

**Soundness:** 3 good
**Presentation:** 3 good
**Contribution:** 3 good
**Rating:** 7
**Confidence:** 3

**Summary:**

This paper propsed a framework to measure model biases by ranking images within their class based on the strength of spurious cues and evaluate the gap in accuracy on the highest and lowest ranked images.
The analysis is comprehensive for a very large number of models.


**Strengths:**

This paper propsed a simple method to measure model biases by ranking images within their class based on the strength of spurious cues.
The proposed method is simple but novel and unprecedented and seems reasonable.
In particular, the most strong point is that the analysis is comprehensive for a very large number of models.


**Weaknesses:**

It seems more compatible with a computer vision conference than a machine learning conference.
The definition of spuriosity in Sec 4.1 is a bit ad hoc.
The ending without discussion or conclusion is a bit of a dead end in terms of the paper structure.


**Questions:**

The definition of spuriosity in Sec 4.1 appears to implicitly assume the normality with respect to r.
For example, if the distribution of r is long-tailed, the variance would be too large and this definition is undesirable.
This may not be a problem if the features are after batch normalization, but for some models, this calculation may b undesirable, and would that affect the measurements?

**Limitations:**

The validity of the interpretation method for deep learning models itself is directly related to the validity of this method.
The significance and meaningfulness of the proposed method is entirely dependent on the  interpretation model.
Because several papers have shown that explanations using heat maps are sometimes fragile, the usefulness of this technique is influenced by the goodness of other techniques.

---

> ### Author Rebuttal · Authors · 2023-08-09
>
> We thank the reviewer for their kind words. We now address each concern.
>
> **On the modality-agnostic potential of Spuriosity Rankings**. While we demonstrate our method on a fundamental task (classification) in a fundamental modality (vision), we believe our framework and the lessons obtained by them are relevant for all in the ML community.
>
> We demonstrate that the tendency of ML models to specialize the function of deep neural nodes can be leveraged to scalably sort data, specifically by finding such nodes that focus on spurious features. We also show the power of sorting data, to various ends, including interpreting spurious features, measuring bias caused by them, and fixing those same biases. We hope our work will inspire people from all subfields to consider sorting their data, namely by re-using existing models.
>
> Our large-scale empirical study also sheds important insight to the spurious correlation problem, which is pervasive in all ML subfields. Namely, we learn that since diverse models seem to inherit the same biases, data-centric approaches are needed. Further, we detail the crucial class-dependence in defining and handling spurious correlations. Since most existing research around spurious correlations (across fields) is model/algorithm-centric and class-agnostic, we hope our findings will be novel and insightful to a wide audience.
>
> While implementation details (i.e. how to interpret and utilize spurious concept detectors inside existing models) will vary, we believe that Spuriosity Rankings at its core is modality-agnostic, and thus potentially impactful to anyone studying spurious correlations.
>
> **Conclusion**. We will add a conclusion in the updated draft (where we will have an extra page) so to tie together the whole work, with emphasis on higher level takeaways that apply to wider audiences, like the ones mentioned above.
>
> **Spuriosity approximation in Sec 4.1**. We note that we offer just one way to approximate spuriosity, which is in many ways qualitative. We provide more detailed explanations of the notion of spuriosity and its implications in the introduction (L30-37). We approximate spuriosity quantitatively by averaging normalized activations of robust neural features annotated as detecting spurious features. We normalize these activations so that one neural feature does not dominate others in the case that feature activation distributions vary significantly (i.e. the long-tailed case the reviewer describes). Moreover, since we ultimately wish to rank images, we aim to average the ‘percentile’ an image falls w.r.t. activation on any of the spurious neural features, which is proxied by the normalized activation we employ. At the suggestion of the reviewer, we inspect activations of neural features in a diverse model set, finding that they all generally follow a normal distribution on their right tail. Further, within each model, activation variances are similar across neural features. Thus, while we thank the reviewer for drawing attention to a potential shortcoming, we feel comfortable in the reliability of our method. Nonetheless, spuriosity can be approximated in different ways, and we are eager to see how others in the community will do so in future work.
>
> **Quality of employed interpretability methods**. We employ an interpretability method with significant precedent [42, 40 in original references], and further conduct extensive human validation of our interpretations (detailed in Appendix F). We agree with the reviewer that the success of our approach relates to the quality of interpretability methods employed, and we argue that this is a strength, as our method will continue to improve as models become more interpretable, which due to increased regulation, we believe is a certainty in the future.

---

### Official Review · Reviewer_ck2C · 2023-07-06

**Soundness:** 3 good
**Presentation:** 4 excellent
**Contribution:** 3 good
**Rating:** 6
**Confidence:** 4

**Summary:**

The paper proposes a method to rank images in a given dataset in order of their "spuriosity". The proposal uses methodology from previous work [42] which examines most-active neural features of a trained model at per-class level, and hand-labels some sample images w.r.t. whether those features are “core” or “spurious”.

The authors 1) scale out the labeling to imagenet-1k, 2) use these sample labels to rank instances according to "spuriosity", 3) present a range of qualitative and quantitative results arguing that they can, e.g., reveal minority subpopulations, measure model bias & mitigate it, enable study of model stability by distributional perturbations, identify noisy labels, etc.

**Strengths:**

1. The authors give a data centric view to the problem of spurious correlation – i.e., instance rankings instead of feature / attribute labels or descriptions. This paper extends the study of [42] for all 1000 classes of imagenet.
2. The authors present some useful analyses e.g., showing ubiquity and correlation of bias across models and training methods, demonstrating “auto-”discovery of novel dimensions of bias using the new supervision, and  the class-specificity of certain biased features. In particular, it makes sense, as the authors argue, to consider the “spuriosity” of a feature at class granularity rather than dataset granularity.
3. The proposal of data rankings opens up the possibility of trying out many mitigation methods and metrics to evaluate bias. The authors demonstrate one simple mechanism, and also suggest that their rankings can help flag mislabelled examples.


**Weaknesses:**

1. There is no quantified measure to evaluate the quality of rankings proposed. Is it possible to look at synthetic or small data to get a quantitative sense of ranking quality?
2. The bias mitigation approach is not compared with the very wide existing literature on bias / spurious feature mitigation – this limits a thorough assessment of  the value of the proposed rankings in mitigation.
3. The claim about identifying incorrect labels is largely qualitative – some quantification would strengthen the case.
4. The process of identifying spurious correlations requires human supervision–this can limit the applicability of the approach, both in terms of only considering or covering some aspects of bias, and in terms of labeling costs which are higher than typical category labeling.


**Questions:**

1. Overall, if many of the qualitative / exemplar applications could be quantified with more rigorous comparisons, this would strengthen the paper. E.g., recovery of “known” rankings, human-rating or other quantification of ranking quality, comparison against at least some simple bias mitigation methods,  quantitative success at identifying label noise, etc.
2. Are the rankings / human labeling data to be released publicly? Are the authors claiming a “dataset contribution” as part of their submission?


**Limitations:**

The process of identifying spurious correlations depends on a) neural activations of trained models, and b) human supervision–this can limit the applicability of the approach, both in terms of only considering or covering some aspects of bias, and in terms of labeling costs which are higher than typical category labeling (although, of course, the entire dataset does not need to be labeled).

---

> ### Author Rebuttal · Authors · 2023-08-10
>
> We thank the reviewer for taking the time to truly engage with our paper and provide insightful comments. We now address main concerns.
>
> **Validating Rankings**. Given the complexity of sorting thousands of images, evaluating the quality rankings is non-trivial, especially since a single ground-truth ranking often does not exist. Nonetheless, we validate several key properties of our rankings. First, we claim our rankings organize data along interpretable notions. Prior work shows that robust neural features are interpretable [42], and we similarly confirm this for all robust neural features used in our study (Appendix F.1.1). Next, we claim that our rankings identify potential biases caused by spurious correlations. In a large-scale evaluation of 89 models, we find that all models underperform on low spuriosity images, thus validating that our rankings capture biases. Further, we claim that high spuriosity images contain the spurious concept of interest. We validate this with a human study, finding that images in the top 20th percentile of spuriosity contain the intended spurious cues in nearly 90% of cases (see Appendix F.1.2). To provide further validation, we conduct another validation in our rebuttal. To confirm that the low spuriosity images do not contain the relevant spurious cues, we inspect the ten lowest spuriosity images for 20 randomly selected classes. We find that the relevant spurious cues are absent in 97% of inspected training images and 93% of inspected validation images. See Appendix C for a visualization of 5 randomly selected classes.
>
> While the specific rankings obtained from our proposed framework do not have a straightforward evaluation procedure, this does not take away from the value of the framework as a whole. We believe the core idea of scalably sorting data along interpretable directions by leveraging existing models is novel, significant, and effective, as demonstrated by our experiments. We thank the reviewer for recognizing that our proposal ‘opens up the possibility of … many mitigation methods and metrics.’
>
> A Potential Automated Quantitative Scheme to Validate Rankings
> Despite the challenging nature of validating rankings, we believe an automated method can be constructed. Namely, one could utilize an open-vocabulary object detector or image tagger to annotate the objects / concepts that are present in highest and lowest spuriosity images (based on some proposed ranking we wish to evaluate). Then, one could assess ranking quality as the difference in presence of some detected spurious tag between the most and least spuriosity images. We find this an interesting avenue of future research, but opted for human validation in this paper, so as to avoid inheriting biases from the auxiliary model used in validation. We agree with the reviewer that such an automatic quantitative evaluation method would be of value, and hope to pursue this in follow up work.
>
> **Comparing to existing bias mitigation techniques**. We prioritized conveying our approach over comparing to other methods, many of which differ in fundamental ways (e.g. data centric vs agnostic) from our idea. Nonetheless, we agree with the reviewer that more comparison would be insightful. We'll include more discussion of other methods in the revised draft.
>
> Further, we perform a new experiment in the rebuttal period, in which we finetune model classification heads on misclassified samples (a common approach). We find that finetuning on misclassified training samples does not close spurious gaps. Also, tuning on errors leads to a larger reduction in validation accuracy (see figure in attached pdf). We conjecture this occurs because errors can be caused by many things aside from the absence of spurious cues, such as label noise. Thus, tuning on misclassified samples could lead to overfitting on unreliable data. In contrast, low spuriosity images are designed to only differ from typical samples in that they lack particular cues that have already been deemed spurious by a human. Arguably, low spuriosity images can at times offer more reliable learning signal, as they do not contain distracting shortcuts (e.g. the low spuriosity lighters in figure 1 are far easier to see than the high spuriosity ones).
>
> **Validating label-noise flagging**. We thank the reviewer for reading our paper to the end! In section 5, we present qualitative evidence that spuriosity rankings can aid in flagging mislabeled samples, as high spuriosity images with severe negative gaps appear to be mislabeled. To strengthen this claim, we conduct a new experiment leveraging ImageNet ReaL labels [1], which indicate for each ImageNet validation image whether objects from multiple classes are present. ReaL labels show 14% of validation images contain multiple objects. We find that for classes with the 5 most negative spurious gaps (averaged over our model suite), high spuriosity (i.e. ranked in top 10% of spuriosity) validation images contain multiple objects in **80%** of cases. In contrast, low (bottom 10%) spuriosity images from the same classes contain multiple objects in only 8% of cases, indicating that the label noise is likely a result of the hypothesized spurious feature collision. Similarly, for classes who’s spurious gap is less than -20%, ReaL labels reveal 42% of the high spuriosity images to contain multiple objects, many times higher than the average rate of label noise. We'll include discussion of this quantitative validation in the updated draft and thank the reviewer for the suggestion.
>
> **Release**. We cannot recall if a dataset contribution was claimed, though regardless, we intend on taking every step to keep our work reproducible and accessible to future researchers. We have already built a web UI for easy viewing and access of our annotations.
>
> **Human cost**. See global rebuttal for ideas on how Spuriosity Rankings can be fully automated.
>
> [1] Are we done with ImageNet? Beyer et al, 2020

---

> > ### Comment · Reviewer_ck2C · 2023-08-13
> > **response to authors**
> >
> > Thank you for your detailed response.
> >
> > I remain convinced that the work is broadly along directions useful to image understanding and machine learning in general.
> >
> > I am comfortable with my assessment / rating of "weak accept".

---

### Official Review · Reviewer_4unL · 2023-07-07

**Soundness:** 3 good
**Presentation:** 2 fair
**Contribution:** 3 good
**Rating:** 6
**Confidence:** 3

**Summary:**

- This work proposed Spuriosity, a quantity for determining the spuriosity ranking of data. The framework builds upon [42], which identifies spurious and core neural features by analyzing the neural activation map of an adversarially trained model. Spuriosity is defined based on these spurious neural features. The author also proposed the concept of ‘spurious gap,’ which measures the accuracy drop between the top-k highest and lowest validation images. By re-training the classifier head from a subsampled dataset using spuriosity, the authors demonstrate a reduction of around 10-20% in the spuriosity gap at the cost of 1-3% reduction in validation accuracy.
- While I acknowledge the challenging nature of the addressed problem and the novelty of this work, I believe that the experiment in this paper primarily focuses on the use case of the proposed quantities rather than proving comprehensive validation of them.

**Strengths:**

- This paper addresses the challenging problem: figuring out the spuriosity of large-scale datasets and investigates its impact on model training. I highly appreciate these contributions, as the field often focuses on improving the performance of simplistic benchmark datasets such as Waterbirds.
- The paper introduced a novel approach that leverages the neural features to distinguish minority samples, which is algorithmically distinct from the previous works such as [1, 2, 3], which rely on classification results of auxiliary models.
- Their method can be effectively combined with balanced re-training methods, such as DFR.


[1] Liu, Evan Z., et al. "Just train twice: Improving group robustness without training group information." International Conference on Machine Learning. PMLR, 2021.

[2] Kim, Nayeong, et al. "Learning debiased classifier with biased committee." Advances in Neural Information Processing Systems 35 (2022): 18403-18415.

[3] Hendrycks, Dan, et al. "Natural adversarial examples." Proceedings of the IEEE/CVF Conference on Computer Vision and Pattern Recognition. 2021.

**Weaknesses:**

- Regarding the Spuriosity ranking, I have concerns about the lack of quantitative metrics and comparison with baselines, making it hard to estimate how reliable the suggested quantity is and how much better compared to the other methods. I suggest comparing the performance of detecting minority samples with [1, 2] on the group annotated benchmark datasets such as WaterBrids and CelebA.
- Another concern is the lack of a baseline and metrics of Spurious Gap. Say, someone proposed a Spurious Gap ver.2, then how to know which one is better?
- The proposed method for detecting spurious correlation relies on human effort.
- Minor correction for misinterpreted terminologies
    - Interpretable model: Consider using an ‘adversarially trained model,’ as normally trained models are also interpretable.
    - [L239] How it is trained: This expression is too broad. It seems to encompass algorithmic approaches like GroupDRO, etc.

**Questions:**

- [Figure 8] How about comparing the reduction in the spurious gap with a subsampled dataset constructed using misclassified data from an auxiliary model?
- Is there any idea for establishing the quantitative metrics and baselines of Spuriosity ranking and Spurious gap?

**Limitations:**

- Their method heavily relies on interpretation methods and human judgment, which is costly.
- The utilized neural activation map cannot distinguish certain features that are not discernible based on spatial information alone, such as the color and texture of the objects.
- Most of the framework originates from previous work [42], which diminishes the contribution of this paper.

---

> ### Author Rebuttal · Authors · 2023-08-10
>
> We sincerely thank the reviewer for taking the time to read our paper and provide insightful comments. We address them below.
>
> **Comparison to existing baselines**
> We thank the reviewer for noting that our method tackles the “challenging problem [of] figuring out spuriosity on large scale datasets”, in contrast to most work which “focuses … on simplistic benchmark datasets”. We also thank the reviewer for their suggested follow up experiment, in which we finetune model classification heads on misclassified samples, a la JTT or LfF. To provide additional quantitative demonstration of the advantages of our approach, we perform this experiment, and find that finetuning on misclassified training samples does not close spurious gaps. Also, we find that tuning on misclassified samples leads to a larger reduction in validation accuracy (see table attached to global rebuttal). We believe the reason for this is that misclassifications can be caused by many things aside from the absence of spurious cues, including label noise. Thus, tuning on misclassified samples could lead to overfitting on unreliable data. In contrast, low spuriosity images are designed to only differ from typical samples in that they lack particular cues that have already been deemed spurious by a human. Arguably, low spuriosity images can at times offer a more reliable learning signal, as they do not contain distracting shortcuts (e.g. the low spuriosity lighters in figure 1 are far easier to see than the high spuriosity ones).
>
>
> **On evaluating ranking quality, with a potential automated solution**
> We thank the reviewer for their important question. We note that given the complexity of sorting a large number of images, validating/evaluating the quality rankings is non-trivial, especially since a single ground-truth ranking often does not exist. In our work, we primarily use human studies to validate several key properties of our rankings; we refer the reviewer to our rebuttal to Reviewer ck2C for more details.
>
> While we believe our validation is convincing, we agree that in the future, an automated quantitative procedure would be valuable to the community. In response to the reviewer’s question, we hypothesize the following scheme: one could utilize an open-vocabulary object detector or image tagger to annotate the objects / concepts that are present in highest and lowest spuriosity images (based on some proposed ranking we wish to evaluate). Then, one could assess ranking quality as the difference in presence of some detected spurious tag between the most and least spuriosity images. Options for quantitative metrics could be the difference in (a) number of times a spurious concept occurs in the the top vs. bottom spuriosity images, (b) similarity of top vs. bottom spuriosity images to the text embedding of a spurious concept (embedded using a vision-language model like CLIP). We find this an interesting avenue of future research, but opted for human validation in this paper, so as to avoid inheriting biases from the auxiliary model used in validation. We agree with the reviewer that such an automatic quantitative evaluation method would be of value, and hope to pursue this in follow up work.
>
> **Minor corrections**. We thank the reviewer for the detailed comments. We will use softer language in L239. We note that we intentionally use the term ‘interpretable’ model, as our overall framework can certainly extend to non-adversarially trained models. So long as one can interpret internal nodes or, more broadly, directions in the representation space of a deep model, one could use the model to extract spurious concept detectors and proxy spuriosity quantitatively.
>
> **Use of human judgment**. We refer to the global rebuttal, where we detail how human involvement can be removed if desirable.
>
> **Beyond spatial interpretation of neural feature focus**. While NAMs only highlight a particular image region, feature attacks change the image to amplify the true concept causing activation on the neural feature of interest. For example, in Figure 3 (right), while the NAM may leave question as to whether the butterfly or flower causes activation, the feature attack actually replaces the butterfly with more flowers, making the focus of the feature crystal clear.
>
> **Contributions over prior work**. We note that the key idea of our work (i.e. sorting data at scale towards interpreting, measuring, and mitigating bias) is novel relative to the last work. While we utilize the spurious feature discovery method of [42], we believe it is just an implementation for one step of the pipeline (extracting concept detectors from trained networks). We refer to Appendix F.3 for extensive discussion of the novel contributions of our work compared to [42].

---

> > ### Comment · Reviewer_4unL · 2023-08-20
> > **Response to authors**
> >
> > I appreciate your detailed response.
> >
> > I believe the contributions of this paper are good enough to be accepted. However, there are remaining concerns.
> >
> > - As y13p mentioned, revising the paper to be self-contained would be important.
> > - I agree that the result presented in Appendix F.1.2 is convincing for validation. Nonetheless, I believe there are still remaining tasks required to fully validate the Spuriosity ranking. For example, within this paper (https://arxiv.org/abs/2206.10843), Figure 4 presents the ability to rank the spuriosity of the training dataset. Also, one could establish the naive ranking based on learning speed (sorting samples based on training loss). Although these methods might be affected by label noise, employing them as baselines and checking their performance through the human study in Appendix F.1.2 could benefit this community.
> >
> > Many of my original concerns are alleviated now, so I increased my review score.

---

### Author Rebuttal · Authors · 2023-08-10

We sincerely thank all reviewers for their time and insightful comments. A couple reviewers wonder if the involvement of a human in our framework is a limitation. We provide extensive discussion below, where we argue that having a human in the loop is a strength, though it is not necessary, as automated alternatives are completely feasible within our proposed Spuriosity Rankings framework.

We would also like to draw attention to the four small additional small experiments/validations conducted during the rebuttal period.
i. We add a new baseline inspired by a common family of approaches in bias mitigation, where a model upweights error samples. We find that this tuning does not close spurious gaps and reduces validation accuracy more than our proposed method.
ii. We perform a small scale human study to validate that low spuriosity images do not contain the relevant spurious cues detected by robust neural features annotated as spurious.
iii. We leverage ImageNet ReaL labels to quantitatively verify that the rate of label noise is significantly higher amongst high spuriosity images from classes with strong negative spurious gaps, as mentioned in Section 5.
iv. We demonstrate strong preliminary signal that an automated version of Spuriosity Rankings leveraging recent VLMs and LLMs can very efficiently extract interesting model biases (details below).

We are grateful that the reviewers see the merits of our simple yet powerful idea to sort existing data using existing models so that we can better utilize them, towards more reliable/less biased ML models. We hope our rebuttals have addressed most of the reviewer comments. Thank you.

**On human involvement**
We believe that human involvement is a strength of our framework, as it increases transparency. Namely, the human in the loop is given a concise inside look on the cues a model trained on the given data is likely to rely upon. Moreover, the human is given agency to decide which cues model performance should be invariant to. Next, we note that the level of human involvement is relatively low. Given the complexity of the task of sorting thousands of images within a class, we believe that only requiring a human to inspect a handful of images is quite efficient. With an appropriate UI, we can confirm that this takes no longer than about a 30-40 seconds per class.

However, in cases where minimizing cost is preferred, Spuriosity Rankings can absolutely be automated. One way to do so is to automate the annotation of neural features as core or spurious. Specifically, one could automatically segment the class object with open-vocabulary segmentation models [1], and then compute the amount of saliency placed on the object by the neural activation map. If most of the salient pixels for a neural feature lie outside of the image region containing the class object, one could automatically flag such a feature as spurious. Indeed, a similar idea is explored with promising early signs in this recent workshop paper [2].

Another way to automate Spuriosity Rankings is to leverage vision-language models (VLMs) like CLIP [3]. With VLMs, we can compute the similarity of an image to any concept, encoded directly from text. After sorting images by similarity to spurious concepts, we can inspect the Spurious gap to measure bias, and train on low spuriosity images to mitigate bias. Further, one can utilize an LLM to automatically generate spurious concepts that a model may rely upon. To demonstrate the feasibility of this extension, we perform it to discover spurious correlations for CLIP. Specifically, we ask Vicuna-13b ‘List 16 different locations in which a {classname} may appear in an image’ and 'In an image of a {classname}, list 16 other objects that may also appear' for all classes in ImageNet, so to extract common backgrounds and co-occurring objects (i.e. potential relevant spurious cues). We then rank images within their class by spuriosity (measure by similarity of image embedding to the text embedding of ‘a photo of a {LLM inferred potential spurious concept}’), compute spuriosity gaps, and inspect class-concept pairs with the highest gaps. Indeed, the framework detects numerous interesting biases, such as ‘clay’ for the class potter’s wheel and ‘credit cards’ for the class wallet. Moreover, inspection of the images with most and least spuriosity confirm that Spuriosity Rankings computed in this manner also can extract from one’s own data natural examples and counterfactuals of the spurious correlation (e.g. potter's wheel images with and without clay). See the attached pdf to inspect these images.

Note that the underlying mechanism of Spuriosity Rankings is preserved: we utilize the representation space of an interpretable model to quantify the presence of relevant cues, and rank images by spuriosity (proxied by similarity/activation of concept directions in representation space) to enable interpretation of spurious cues in context and measure model bias caused by them. While we believe the human in the loop of our presented framework offers some advantages, we appreciate the reviewers' questions regarding automated alternatives, are confident that they exist, and will include some discussion of them in the revised draft.

[1] Segment Anything, Kirillov et al, 2023
[2] Identifying and Disentangling Spurious Features in Pretrained Image Representations, Darbinyan et al, 2023
[3] Learning Transferable Vision Models from Natural Language Supervision, Radford et al, 2020

---

### Decision · Program_Chairs · 2023-09-21

**Decision:**

Accept (spotlight)

**Comment:**

The paper suggest a new method for measuring spuriosity of data examples, namely ranking data points by the spurious features that they contain. To do this, the authors propose to use a method combining adversarial training, utilizing deep features indicating spurious correlations and manual labeling. This results in a method for sorting the data examples from low to high spuriosity. The authors show that this can be used to mitigate problems that arise from training on data with spurious correlations, identify minority subpopulations and combat class-wise bias.

It seems that after discussion with the authors, all the reviewers are supporting the paper for acceptance. Some of the reviewers were concerned about the requirement of using human feedback in the spuriosity evaluation framework, but to my understanding the response of the authors answered most of the concerns.

I will therefore recommend that the paper is accepted, and urge the authors to improve the paper based on the comments raised by the reviewers.